# Salivary Gland Bioengineering

**DOI:** 10.3390/bioengineering11010028

**Published:** 2023-12-26

**Authors:** Stephen C. Rose, Melinda Larsen, Yubing Xie, Susan T. Sharfstein

**Affiliations:** 1Department of Nanoscale Science and Engineering, College of Nanotechnology, Science, and Engineering, University at Albany, SUNY, 257 Fuller Road, Albany, NY 12203, USAyxie@albany.edu (Y.X.); 2Department of Biological Sciences and The RNA Institute, University at Albany, SUNY, 1400 Washington Ave., Albany, NY 12222, USA; mlarsen@albany.edu

**Keywords:** salivary gland, organoid, tissue engineering, regeneration

## Abstract

Salivary gland dysfunction affects millions globally, and tissue engineering may provide a promising therapeutic avenue. This review delves into the current state of salivary gland tissue engineering research, starting with a study of normal salivary gland development and function. It discusses the impact of fibrosis and cellular senescence on salivary gland pathologies. A diverse range of cells suitable for tissue engineering including cell lines, primary salivary gland cells, and stem cells are examined. Moreover, the paper explores various supportive biomaterials and scaffold fabrication methodologies that enhance salivary gland cell survival, differentiation, and engraftment. Innovative engineering strategies for the improvement of vascularization, innervation, and engraftment of engineered salivary gland tissue, including bioprinting, microfluidic hydrogels, mesh electronics, and nanoparticles, are also evaluated. This review underscores the promising potential of this research field for the treatment of salivary gland dysfunction and suggests directions for future exploration.

## 1. Introduction

In the United States, over twenty-four million people suffer from xerostomia, or dry mouth, with over a million people experiencing moderate to severe symptoms [1]. More than 400 medications cause xerostomia; elderly individuals suffer disproportionately from xerostomia as a result [2]. Other medical conditions such as Sjögren’s disease and diabetes also cause xerostomia. Seventy-four to eighty-five percent of head and neck cancer patients receiving radiation therapy experience xerostomia [3]. According to World Cancer Report 2014, there were 550,000 head and neck cancer cases worldwide, suggesting the generation of at least 351,000 new cases of xerostomia from head and neck cancers alone [4]. The global xerostomia therapeutics market size was estimated at USD 625.3 million in 2018 and is expected to grow at a CAGR of 3.6% between 2019 and 2026 [5].

Xerostomia, which is the perception of dry mouth, typically results from a defect in saliva production or secretion, which is known as hyposalivation. Hyposalivation leads to a significant decrease in quality of life as it damages oral and general health, causing cracked lips, microbial proliferation, periodontitis, cavities, oral ulcerations, and difficulty in speaking, eating, tasting, swallowing, and digesting [6]. Poor oral health can lead to compromised oral barrier function resulting in the passage of bacteria. As a result of an oral barrier compromise and the subsequent systemic inflammation related to bacterial pathogens and endotoxins, poor oral health can contribute to cardiovascular and related diseases and is associated with rheumatoid arthritis, Alzheimer’s disease, cardiovascular, diabetes, and cancer [7]. Currently available treatments include medication changes, saliva stimulants (sialagogues), and moisturizers; however, these treatments offer only transient relief and sialagogues can have side effects, including sweating, increased heartbeat, abdominal pain, diarrhea, nausea, and changes in vision, among others. Temporary relief is a suboptimal solution, particularly for many head and neck cancer survivors experiencing persistent xerostomia due to the limited salivary gland regenerative capacity [8].

Several approaches may improve the regenerative capacity of salivary gland tissues including gene therapy, stem/progenitor cell-based therapy, and tissue engineering strategies, many of which are nearing clinical translation. Additionally, advances in organoid technology, organoid-enabled discovery of therapeutic targets, drug screening and testing, and tissue regeneration will provide new avenues to salivary gland tissue regeneration. This article will first review salivary gland biology and pathologies driving the development of regenerative medicine methods to restore salivary function and then review progress towards achieving regenerative medicine-based strategies for clinical application, followed by future perspectives on next generation salivary gland tissue engineering and regenerative medicine.

## 2. Learning from Salivary Gland Biology and Pathology

### 2.1. Salivary Gland Development and Function

#### 2.1.1. Salivary Glands and Their Functional Units

Salivary glands function to produce a composite of water, electrolytes, enzymes, and mucus called saliva [9]. Ninety percent of saliva is produced by three sets of major salivary glands: parotid, submandibular, and sublingual (Figure 1) [10]. Gland secretions vary between the glands, with the parotid glands in front of and beneath the ear producing primarily the watery and proteinaceous components of saliva, the sublingual glands (SLGs) beneath the tongue primarily producing mucus, and submandibular glands (SMGs) on both sides, just under the jaw, producing both components [11].

These salivary glands are innervated by the autonomic nervous system. Parasympathetic innervation, via the cranial nerves, tends to promote saliva secretion. The glossopharyngeal nerve innervates parotid glands, and the facial nerve innervates SMGs and SLGs [12]. Additionally, hundreds of minor salivary glands line the surface of the mouth and throat. Each gland is divided into lobules; these lobules in turn are made up of functional units called adenomeres. The adenomere contains several cell types: acinar (serous and mucous), myoepithelial, basal, and ductal (intercalated, striated, and excretory) cells. The adenomere does not function independently, but rather in an integrated fashion with neighboring parasympathetic ganglia and blood vessels [11,13].

#### 2.1.2. Coordination of Saliva Secretion

Both the adenomere and the acinus have been referred to in the literature as the functional unit of the salivary gland. The distinction between the two is that the adenomere also includes the intercalated, striated, and excretory ducts, which provide the necessary conduits for saliva expulsion into the oral cavity in addition to the acinus. In each adenomere, innervated myoepithelial cells surround clusters of acinar cells, each arranged in a bulb-shaped formation (Figure 2). Serous acinar cells secrete a watery fluid devoid of mucus; mucous acinar cells produce a secretion rich in mucins. Mucins are glycoproteins that combine with water to form a mucin hydrogel (mucus). The secretory products of each pair of glands are governed by the distribution of serous acinar versus mucous acinar cell types. Accordingly, the parotid gland, known as a serous gland, releases a watery secretion contributing to around 25% of our daily saliva. Its secretions enter the oral cavity via ducts situated in the cheek lining, aligning roughly with the molar teeth. SMGs, located on the lower jaw, produce both watery and mucous secretions, contributing to a significant 70% of total salivary output. Their secretions flow into the mouth through ducts positioned on both sides of the tongue’s base, near the thin frenulum. In contrast, the SLG produce primarily mucous secretions in smaller quantities, dispensed through several tiny ducts beneath the tongue [14]. It is worth noting that in humans, a given acini can be mixed (seromucous acini) and in mice, the SMG acini contain seromucous cells [15].

Production of saliva requires the coordinated efforts of multiple cell types. Myoepithelial cells contain actin filaments, which contract to increase hydrostatic pressure in neighboring cells. This myoepithelial cell contraction results in the expulsion of acinar-cell secretory products into the ductwork of the adenomere [16]. Acinar secretions are deposited into the ductal system, initially entering through the intercalated duct. Acinar and ductal cells are linked together by multiple classes of complementary cell junctions such as occluding, anchoring, and communicating junctions, which maintain the structural and mechanical integrity of the adenomere structure [17]. Together with the basement membrane, the myoepithelial cells surround the acini and line the abluminal layer of the intercalated duct. The lumen of the intercalated duct is continuous with the striated duct. The characteristic appearance of the striated duct results from numerous folds in the apical plasma membrane that facilitate ion exchange to modify the acinar cell saliva. The apical domain of the striated duct cells secretes HCO_3_^−^ and K^+^ and reabsorbs Na^+^ and Cl^−^ using the Na^+^-K^+^ pump and the Cl^−^-HCO_3_^−^ pump to make the saliva hypotonic. This would not be the case were it not for the ductal cells being relatively impermeable to water. This causes the water in the saliva to remain in the duct and not be reabsorbed back into the bloodstream, and thus, the saliva becomes diluted or hypotonic compared to plasma by the time it exits the ducts into the oral cavity [11]. Further, the overall reabsorption of Na^+^ and Cl^−^ is higher on a molar basis than the secretion of K^+^ and HCO_3_^−^, which further contributes to the hypotonic nature of the saliva.

The production of saliva requires the apicobasal polarity of the epithelial cells, which develops following epithelial cell–cell interactions and interactions with basement membrane proteins [18] that are produced by both epithelial and stromal cells [19].

#### 2.1.3. Salivary Gland Development

##### Branching Morphogenesis Creates the Adult Gland Structure

Throughout nature, branching morphogenesis serves to form complex, well-ordered architectures. This meticulous formation arises from the cellular and molecular coordination that sets the stage for the unfolding of tissue development. Paramount in this orchestration are the molecular cues that navigate the dynamic cellular landscapes, ensuring their timely and spatial responses.

Fibroblast growth factors (FGFs), especially FGF7 and FGF10, act as liaisons between the mesenchyme and epithelium. Their paracrine signaling is pivotal for driving epithelial proliferation and, in turn, shapes the eventual branching architecture of the gland [20,21]. Other morphogens, such as bone morphogenetic proteins (BMPs) and pivotal members of the transforming growth factor β (TGF-β) superfamily, orchestrate cellular proliferation and differentiation [22]. Their nuanced gradient distributions within developing tissues help establish unique cellular zones, dictating specific branching patterns within the salivary gland [23]. Not to be overshadowed, the Sonic hedgehog (Shh) signaling pathway, central in many developmental processes, carves its niche particularly in the early stages of salivary gland morphogenesis [24].

Moving to the cellular adhesion process, cadherins emerge as central players. These calcium-dependent adhesion proteins ensure epithelial cell cohesion, with E-cadherin playing a lead role [25]. Its dynamic modulation during morphogenesis facilitates the flexible cellular rearrangements necessary for bud formation and clefting processes, as reviewed by Sisto et al. [25]. Integrins, the bridges between the cells and their extracellular matrix (ECM), transmit vital signals that inform cellular behavior. Their nuanced interactions with the ECM components, notably fibronectin and laminins, play a cardinal role in the epithelial invagination and migration tales that underscore branching morphogenesis [26].

As the story unfolds, these structures develop from a series of reciprocal interactions between tissue types anchored in conserved molecular programs [18]. This process includes coordinated cellular actions, such as proliferation, ECM remodeling, differentiation, migration, and apoptosis. In the salivary gland, branching morphogenesis functions to maximize surface area, ensuring the flow of saliva from the salivary gland. Branching morphogenesis maximizes the surface area for the secretion of fluid across its underlying epithelium. Normal salivary gland function is dependent on branching morphogenesis of the epithelium to form the parenchymal tissue as well as the coordinated branching of the vasculature and nerves that support and innervate the gland, respectively [20,21].

##### Epithelium, Endothelium, and Neural Crest-Derived Cells Regulate Early Salivary Gland Development

The orchestration of salivary gland development is a testament to the high-level interplay between diverse cell types. Each cell type not only carves its niche but also communicates actively with its neighbors to shape the mature gland. In coordinated unfolding of salivary gland development, the mesenchymal condensates and the thickening process of the oral epithelium precede formation of the primordial bud, or bud initiation, in morphogenesis. The mesenchymal condensates, clusters of mesenchymal cells, prominently express characteristic proteins such as FGF10 and FGFR2b [20,27].

At the same time, as the oral epithelium thickens prior to bud formation, it exhibits a distinctive set of markers. E-cadherin marks the expanding adhesive interactions within this layer [28,29,30]. The epithelial cells initiate the very buds that subsequently undergo branching morphogenesis, setting the stage for the gland’s elaborate architecture. This budding and subsequent branching rely on a myriad of signaling pathways, including those activated by growth factors such as FGFs and epidermal growth factors (EGFs) [31,32]. Proteins such as SOX9, SOX10, and Pax-9 indicate evolving lineage commitments, steering the epithelial cells towards their predetermined roles in the salivary gland development [29,33]. Endothelia form the innermost lining of the blood vessels. Their intricate web ensures the delivery of essential nutrients and oxygen to the developing gland. Furthermore, the endothelial cell signaling can influence the behavior of nearby epithelial and mesenchymal cells, potentially impacting gland morphogenesis [33].

Neural crest-derived mesenchyme cells originate from the neural crest—a transient structure during embryonic development—and migrate to various parts of the embryo. In the context of the salivary gland, they provide crucial signals for epithelial morphogenesis. They secrete factors that guide the epithelial branching and possibly influence the differentiation of specific cell types within the gland [32].

Lineage tracing experiments have been instrumental in understanding the origins of various cell populations in the gland. The Sox17-2A-iCre/R26R experiments highlight the ectodermal origin of major salivary glands, emphasizing the shared developmental lineage between the skin, nervous system, and the glands [31]. This underscores the deeply evolutionarily conserved developmental pathways shared among various organs. Among the factors governing differentiation and lineage commitment in salivary gland development, the SOX transcription factors, particularly SOX2, SOX9, and SOX10, have emerged as pivotal regulators [30,34,35].

SOX2 is a master regulator of the acinar cell lineage. It is expressed in progenitors that give rise to both acinar and ductal cells. However, in its absence, the formation of acinar cells is notably affected, while ductal formations remain largely unaffected. Peripheral nerves further play a role in acini formation through SOX2 regulation [34]. SOX9’s significance in SMGs is underscored by its changing expression patterns from the embryonic to the adult stage. A lack of SOX9 during the early developmental phases leads to smaller initial SMG buds. Additionally, studies have shown that introducing SOX9 into mouse embryonic stem cell-derived oral ectoderms can stimulate salivary gland rudiment development. ChIP-sequencing studies further reinforce the role of SOX9 in guiding genes involved in tube and branching formation [30].

Lastly, SOX10 has proven crucial for both the maintenance and differentiation of specific epithelial progenitors in exocrine glands. Notably, these progenitors are marked by the KIT/FGFR2b/SOX10 axis, representing the earliest multi-potent and tissue-specific progenitors of exocrine glands. When SOX10 is genetically deleted in the epithelial context, there is a marked loss of secretory units, subsequently reducing organ size and function. However, intriguingly, the ductal tree persists. In the absence of SOX10, the remaining duct progenitors demonstrate a lack of adaptability and cannot properly form secretory units. Yet, when SOX10 is overexpressed in these ductal progenitors, there is an enhancement in their adaptability towards KIT^+^ progenitors, driving the differentiation into secretory units. Thus, SOX10 emerges as a central regulator of plasticity and multi-potency in epithelial KIT^+^ cells across several secretory organs, encompassing the mammary, lacrimal, and salivary glands [35].

Collectively, while the distinct roles of SOX2, SOX9, and SOX10 are becoming clearer in the context of salivary gland development, there remains much to be explored about their interplay, regulatory mechanisms, and potential therapeutic implications.

The fact that parotid glands, SLGs, and SMGs arise from the oral epithelium sheds light on the potential shared molecular mechanisms between the salivary glands and other oral structures [36]. Meanwhile, the revelation from Wnt-1-cre lineage tracing, that mesenchymal and nerve cells in the gland originate from the neural crest, underscores the neural crest versatility and its profound influence on craniofacial development [37].

By mouse embryonic day 11 (E11) or human embryonic day 30 (H30), neural crest cells migrate to the oral epithelium, initializing a mesenchymal condensation (see Figure 3) [38]. This process prompts the oral epithelium to thicken into a placode by mouse E11. The placode dives into the mesenchyme, forming an epithelial bud by E12, eventually becoming a primary bud. This action underpins salivary gland branching development.

Simultaneously, nerve connections of the gland are established. Neural crest cells evolve into the parasympathetic fibers for the facial nerve, integrating with the SMG and SLG. By E12, these fibers structure the parasympathetic ganglion (PSG) [39,40,41]. For the gland’s continued maturation at E12/H36, interactions between basal cells, marked by keratin-5 (K5), and the PSG are essential [42]. Wnt signals, emitted by K5^+^ gland cells, regulate this nerve interaction [43].

The parasympathetic ganglion cells also depend on neurotrophic factors such as neurturin (NRTN). Diminishing NRTN in adult mice experiments notably hampers gland regeneration and saliva production. At the molecular level, the glial-derived neurotrophic factor (GDNF) family, inclusive of NRTN, orchestrates nerve cell dynamics. Their receptors, glial cell-derived family receptors α (GFRα 1–4), partner with ligands such as GDNF and NRTN. These ligand–receptor interactions, especially involving NRTN and GDNF, are pivotal for maintaining the health of the SMG’s parasympathetic ganglion. Mechanistically, their association with GFRα and RET co-receptor triggers cellular pathways (e.g., MAPK and PI3K-Akt), overseeing cell growth and survival [44].

Reciprocal Wnt signaling from K5^+^ salivary gland progenitors is necessary for innervation [43]. Wnt signaling, known for its role in cellular proliferation, differentiation, and stem cell maintenance, activates a cascade involving Dishevelled proteins, leading to the stabilization and nuclear translocation of β-catenin, where it can regulate gene expression [45]. In the context of the salivary gland, the specific genes targeted by this signaling might dictate progenitor cell behavior and further differentiation. The progenitors rely on the GDNF family members, such as NRTN. In adult mice, when NRTN and related GDNF family members were experimentally decreased, a 50% reduction in gland regeneration and a 40% reduction in saliva secretion were observed [46].

In general, GDNF family members are involved in survival, proliferation, and differentiation of neuronal populations in the central and peripheral nervous systems. System receptors for these ligands include GFRα 1–4, which are the preferential co-receptors for GDNF, NRTN, artemin (ARTN), and Persephone (PSPN), respectively. NRTN and GDNF are crucial neurotrophic factors with different temporal effects on prenatal and postnatal development and on survival of the SMG parasympathetic ganglion. These effects are mediated through binding to GFRα 1, 2, 3, and the “rearranged during transfection” co-receptor (RET co-receptor), which activates mitogen-activated protein kinase (MAPK), phosphatidylinositol-3-kinase-protein kinase b (PI3K)-Akt, phospholipase c (PLC)-γ, and sarcoma (Src) signaling pathways [44].

In conditions where NRTN and GDNF are experimentally decreased, the salivary gland exhibits marked dysfunction. A significant insight into the protective role of NRTN comes from studies involving irradiation-induced damage in salivary glands. Irradiation, commonly used in the treatment of head and neck cancers, frequently results in irreversible salivary gland hypofunction. Research has shown that NRTN plays a pivotal role in safeguarding the gland against such damage. NRTN assists in the epithelial regeneration of irradiated salivary glands by preventing the apoptosis of parasympathetic neurons, which are crucial for the gland’s function. When delivered via gene therapy, NRTN effectively shields the gland against irradiation damage. Glands pre-treated with NRTN maintain their function post-irradiation, a stark contrast to untreated glands, which exhibit significant dysfunction. This protective mechanism of NRTN operates, at least in part, by bolstering parasympathetic innervation. Markers of parasympathetic function, negatively impacted by irradiation, remain stable when NRTN is present [47].

NRTN’s essential function goes beyond its protective role against external damage like irradiation. In the developing and adult salivary gland, NRTN pairs with GFRα2. While the gland parenchyma produces NRTN, GFRα2 is expressed in the parasympathetic ganglia. NRTN deficiency leads to a substantial loss of GFRα2-expressing parasympathetic neurons in the salivary gland, and the surviving ones appear smaller than in normal conditions. This finding suggests a pivotal role for NRTN as a trophic factor for specific parasympathetic neuron populations within the salivary gland [48].

The epithelium undergoes branching morphogenesis to elaborate its branched structure. At E13/H42, the endbud enlarges, and three to five clefts form in the epithelium as the salivary gland undergoes branching morphogenesis. These clefts later deepen to become the major lobules of the gland. In contrast to other organs, in which branching morphogenesis is driven by proliferation, branching in the salivary gland is driven by the formation of clefts in the basement membrane on the surface of the buds [49]. Cleft formation is initiated by basement membrane fibronectin. Fibronectin drives BTB/POZ domain-containing protein 7 (Btdb7) expression, which induces snail family transcription factor 2 (Snail2) expression and E-cadherin suppression [50,51]. Cleft formation is further reinforced by the action of GSK 3-β, which phosphorylates β-catenin in cells at the base of the cleft, targeting it for degradation. A cytoplasmic shelf with a core of microfilaments occurs in cells at the base of the cleft [52], which may be a matrix attachment point to drive cleft elongation via cytoskeleton attachment. This notion is supported by studies showing that the inhibition of actin cytoskeletal polymerization inhibits cleft formation. Daley et al. proposed that a mechanochemical checkpoint involving rho-associated coiled-coil containing kinase (ROCK) regulates the transition of initiated clefts, which are proliferation independent, to a stabilized state that is competent to undergo cleft progression [53]. Basement membrane is required for cleft formation with laminin α5-null mice showing delayed SMG branching and delayed cleft formation. ROCK also controls organization of the outer layer of cells adjacent to clefts by coordinating cell polarity via PAR-1b protein, which controls positioning of the basement membrane on the basal side of the outer layer of epithelial cells in the developing epithelial buds [54].

Epithelial cell proliferation drives expansion of buds to drive growth of the developing gland. EGFs and their receptors are important for SMG development. EGF-null mice show reduced proliferation, branching, and maturation of the epithelium [55]. Fibroblast growth factors such as FGF1, FGF3, FGF7, and FGF10 are produced from mesenchyme and modulated by platelet-derived growth factor (PDGF) [56]. FGFs are required for salivary gland development as the FGF10- and FGFR2b-null mice do not develop salivary glands [21] and FGF7 produced by myoepithelial cells activates the FGFR2-dependent seromucous transcriptional program to increase saliva secretion and FGF7-FGFR2-MAPK signaling is critical for seromucous acinar differentiation [57]. Proliferation is driven by FGF10 and FGF7, both of which bind to FGFR2b. When FGFR2b binds to FGF10, duct elongation is induced, while binding to FGF7 induces budding [58]. In vivo, FGF10 signaling is likely modulated by heparan sulfate as FGF10 binding to heparan sulfate increases its affinity for FGFR2b, and the ternary complex resulting from this binding increases proliferation [20,59]. Other growth factors are required for salivary gland development, including Wnt and ectodysplasin (Eda), as reviewed previously [38,60,61]. Defining the regulation of different cell populations is critical for the development of regenerative therapies. With recent studies to define distinct cell populations by single cell RNA sequencing (scRNASeq) [61,62], it will be possible to comprehensively define the specific cell populations responding to specific signals.

The adult salivary glands appear to harbor various stem and progenitor cell populations essential for tissue maintenance, regeneration, and repair. Extensive studies have been conducted to pinpoint the exact locations of these cell pools within the salivary glands. The ductal regions, especially the intercalated and striated ducts, are seen as potential storehouses for progenitor cells. Notably, cells that express markers like c-Kit or keratin-5 (K5) reside in these ductal regions, hinting at their stem-like properties [63].

Ascl3, a transcription factor that plays an essential role in determining cell fate, development, and differentiation, marks a progenitor cell population in the adult mouse salivary glands. Ascl3-expressing cells were shown to be intermediate lineage-restricted progenitor cells that exist in all major salivary glands that can differentiate into acinar and ductal cells in vitro in 3D spheres [64].

Basal cells, found at the basal side of the acinar units, are also proposed as potential stem or progenitor cells. They frequently express markers like keratin-14 (K14) and are known to play a part in gland regeneration under certain conditions [65]. Although the primary role of acinar cells is saliva production, they have been observed to dedifferentiate and adopt progenitor-like abilities, suggestive of their ability to aid in glandular repair in certain contexts. Mesenchymal stem cells (MSCs) present in salivary glands are typically found near blood vessels. These cells have the capacity to evolve into multiple cell types, potentially assisting in tissue repair post-injury [66]. In some conditions, salivary gland cells can develop spheroid formations when cultured. These spheroid-forming cells exhibit stem cell traits, positioning them as potential sources for regeneration. In specific mouse studies, Lgr5-positive cells, commonly recognized as stem cell markers in other tissues, have been identified within salivary glands and are suggested to contribute to tissue balance and recovery [67]. It should be noted that the exact roles, traits, and connections of these cell types within human salivary glands remain a subject of ongoing debate.

### 2.2. Fibrosis, Cellular Senescence, and Salivary Gland Pathology

When salivary glands are damaged by immune dysfunction (e.g., Sjogren’s syndrome), radiotherapy, or age-associated cellular senescence, an increase in senescent cells is typically accompanied by fibrosis. In each case, local inflammation is initially or subsequently driven by senescent cells and their secreted products, resulting in inflamed, fibrotic, and senescent cell-enriched salivary glands. In Sjögren’s syndrome, autoimmune dysfunction leads to lymphocyte infiltration of the salivary gland, which is followed by an increase in senescent cells [68]. In salivary glands exposed to radiation, oxidative stress leads to DNA damage, which triggers the DNA-damage response pathways in cells that can lead to cellular senescence or apoptosis [69]. Finally, the number of senescent cells increases with advancing age due to replicative senescence and responses to environmental factors. As senescent cells produce secreted factors as part of their senescence associated secretory phenotype (SASP), senescent cells can affect otherwise healthy cells via bystander effects. This senescence-induced inflammation has many effects that include driving acinar cells to senescence, destroying their ability to produce saliva; dampening or destroying existing stem cells that may exist in or migrate to salivary gland tissue, capping natural regenerative processes; and provoking phenotypic shifts in fibroblasts that drive the unremitting ECM production characteristic of fibrosis [70,71,72]. Numerous studies have shown that the SASP can exert deleterious effects on stem cell function. For example, geriatric stem cells could change reversible quiescence in satellite stem cells into senescence [73]. In the salivary gland, salivary gland stem cells (SGSCs) isolated from Sjögren’s syndrome patients were regeneratively inferior; they were likely to be senescent or limited to intercalated duct cell differentiation only [74].

The impact of radiation on the salivary gland senescence is evidenced by a subtle interplay between different cell types and their physiological responses. A central response is the expression of p16, an inhibitor of cell division kinase 4, seen in the basal cells of the salivary duct, which are believed to act as progenitors. The correlation between p16 expression in these cells with saliva production and the infiltration of CD45^+^ leukocyte cells in Sjögren’s syndrome (SS) patients suggests that basal cell senescence might be an early hallmark of SS, likely contributing to diminished salivary gland function [75]. After radiation exposure, there is a significant loss of acinar cells and shrinkage of the gland during the acute phase [76,77]. In the aftermath of radiation, the resilience and adaptability of different ductal progenitor populations within the gland including KRT14^+^ progenitors is impressive. Fast-cycling cells display an increased proliferation in response to radiation-induced damage and asymmetrically divide to replenish the cells of the larger granulated ducts. On the other hand, KIT^+^ intercalated duct cells are a stark contrast, being long-lived progenitors with minimal divisions both during homeostasis and post-radiation. These cells maintain ductal architecture with slow rates of cell turnover, emphasizing the heterogeneity in response mechanisms employed by salivary progenitor cells to sustain tissue structure [78].

Another facet of radiation-induced effects on salivary glands is their rapid functional impairment, which is evident as soon as 24 h post-exposure. This response underscores the significant impairment of myoepithelial cells, offering a fresh perspective on the pathogenesis of radiogenic salivary gland dysfunction. Beyond just the damage to acinar cells, this indicates that secretory retention, assessed scintigraphically, might be rooted in myoepithelial cell impairment [79,80].

In general, salivary gland cellular senescence and its associated fibrosis create the need for tissue replacement, while simultaneously creating a host tissue environment that is not conducive to it [81].

#### 2.2.1. SASP Signaling Drives Neighboring Proliferation-Competent Cells to Senescence (Bystander Effect)

Senescent cells induce the DNA-damage response in neighboring proliferation-competent cells through a variety of different mechanisms that often results in an increased senescent cell burden. This induction can occur via gap junction-mediated cell–cell contact and processes involving reactive oxygen species (ROS). Continuous exposure induced senescence in bystander fibroblasts [82]. Mikula-Pietrasik et al. showed that senescent human peritoneal mesothelial cells (HPMCs) elicited the bystander effect on neighboring HPMCs and human peritoneal fibroblasts (HPFCs). Further, they identified TGFβ1 as the essential soluble mediator eliciting this change. It was postulated that this effect occurred though the induction of ROS and p38 MAPK. HPMCs also released thrombospondin-1 (TSP-1), a major activator of TGFβ1 [83]. Nelson et al. asserted that ROS-activated NF-κB also elicits the DNA-damage response, leading to senescence in bystander cells [84]. Finally, Da Silva et al. demonstrated the bystander effect in vivo across multiple tissues using NOD SCID gamma mice, which support highly efficient engraftment of human hematopoietic stem cells (hu-CD34^+^) and human peripheral blood mononuclear cells (hu-PBMC) [85]. These findings suggest a possible senescent cell-signaling basis for fibrosis development in salivary glands and decreased saliva production. More importantly, they suggest a set of strategies for mitigating or reversing these effects, which will be explored in a later section.

#### 2.2.2. Senescence and Its Impact on Normal Fibroblast Dynamics in Healing and Fibrosis

Fibroblasts have a set of functionally dynamic phenotypes that are temporally varied according to the healing stage of the tissue in which they reside. Oxidation and other factors associated with wounding prompt fibroblasts to shift their phenotype. While few studies have examined these transitions in the salivary gland, in response to a reversible injury where a metal clip is placed on the primary salivary gland duct, a transient fibrotic response occurs in which Pdgfra^+^, Pdgfrb^+^ fibroblasts overexpress ECM proteins [86]. Reversible phenotypic shifting among fibroblasts is thought to serve as a mechanism for tissue homeostasis [87,88]. In salivary gland organoid culture, Pdgfra^+^ cells that support epithelial cell proacinar differentiation in response to FGF2 can transition to a fibrotic myofibroblast-like phenotype in response to TGFβ1 [88]. In other contexts, these shifts can occur between fibroblast, senescent fibroblast, myofibroblast, and senescent myofibroblast phenotypes. However, it should be noted that fibroblasts may originate from different cell types, which may, in turn, dictate how they shift phenotypically [88,89].

The relationships between senescence and fibroblast dynamics have been explored in other organs and cultured cells. Damaris et al. found that wound-site damage prompted fibroblasts to become senescent and secrete PDGF-AA. PDGF-AA, in turn, prompts wound closure by inducing a phenotypic shift in fibroblasts to myofibroblasts. Contractile elements in the myofibroblasts are responsible for wound closure [90]. In tandem, the myofibroblasts emerge mainly, but not exclusively, as a phenotypic variant of the fibroblasts. Although not all myofibroblasts assume the same function, as a class they produce ECM, close wounds, and release cytokines. IL-10 was shown to play a key role in fibrosis resolution by reversing a TGFβ1-induced, transiently activated, myofibroblast phenotype [91]. These two cytokines can cause neighboring macrophages to become oriented toward ECM degradation, which can lead to the resolution of fibrosis. ECM degradation reduces matrix stiffness, which shifts the apoptosis/senescence axis towards myofibroblast apoptosis. Apoptosis complements the macrophage degradation function to reestablish normal tissue homeostasis. Alternately, myofibroblasts may shift phenotype to become ECM-degrading fibroblasts (deactivation), or they may shift toward senescence as well, by the signaling action of cysteine-rich angiogenic inducer 61 (CYR61/CCN1) [92].

In general, pathological fibrosis is known to occur when myofibroblasts and senescent myofibroblasts escape apoptosis and engage in ongoing overproduction of ECM. Additionally, these cells produce several other compounds that further facilitate pathological remodeling. These pathological changes are readily observed in histological samples of salivary gland tissue that has been damaged due to autoimmune dysfunction, irradiation, or aging. Understanding the factors that control fibroblast dynamics is critical to successful salivary gland tissue engineering. Engineered and host tissue scaffold stiffness and local and systemic signaling aberrations are relevant to transplant success. The same is true of cell-signaling aberrations present in diseased salivary gland and the larger system in which the salivary glands operate. For example, individuals with systemic inflammation will likely have ongoing immune cell infiltration, while individuals with irradiated salivary glands may have different patterns of damage [3]. Aged individuals may have different kinds of cross-linking in their tissue, such as advanced glycation end products, which are less amenable to degradation [93]. Therefore, different patients may require different approaches to salivary gland engineering to optimize transplant and engraftment success.

Scaffold stiffness and growth factors such as TGFβ1 and connective tissue growth factor (CTGF) cue certain fibroblasts to phenotypically shift to the myofibroblast phenotype. This shift is evidenced by the expression of α-smooth muscle actin (α-SMA). ECM stiffness is driven via mechanotransduction pathways that involve lengthy cascades of signaling molecules to control α-SMA transcription. These pathways start at the surface of the cell via β1 integrins. Β1 integrins are connected to F-actin stress fibers, which are bound, in sequence, to FAK, ROCK and myocardin-related transcription factor (MRTF). MRTF can translocate to the nucleus where it binds serum response factor (SRF), which leads directly to the production of α-SMA. MRTF also binds transcriptional co-activators yes-associated protein (YAP) and transcriptional co-activator with PDZ-binding motif (TAZ) to initiate α-SMA transcription. These proteins activate other transcription factors such as TEA domain family member (TEAD), T cell factor/lymphoid enhancer-binding factor (TCF/LEF), and β-catenin [94]. ECM stiffness and mechanical forces also regulate force-dependent activation of latent TGFβ1 by increasing resistance to traction forces generated by fibroblasts. In this mechanism, extracellular latent TGFβ1 (TGFβ1 with its latency-associated peptide) is released from latent TGFβ1-binding protein stores when αV integrins respond to mechanical pulling forces. Once activated, TGFβ1 binds to TGFβ receptors and promotes canonical mothers against decapentaplegic homolog 3 (SMAD3) activation [89,95]. Activated SMAD3 binds to SMAD4 and translocates to the nucleus, where it binds to SMAD-binding elements in the promoters of fibrogenic genes, such as ACTA2 (encoding α-SMA). Together, myofibroblast activation is controlled by both the TGFβ–SMAD pathway as well as biomechanical pathways such as integrin–FAK–ROCK–MRTF–YAP–TAZ signaling [96]. Moreover, ECM stiffness induces expression of the microRNAs miR-21 and miR-29a, which promote the survival of myofibroblasts by increasing the expression of pro-survival BCL-2 proteins [97].

Increased ECM stiffness, TGFβ1 and CTGF all cue fibroblasts to differentiate into collagen-producing, α-SMA^+^ myofibroblasts. These “activated” myofibroblasts also produce the anti-inflammatory cytokines, TGFβ1 and IL-10, which act on macrophages at the injury site, stimulating them to produce ECM degrading compounds. This shift in activity is a part of the resolution phase of normal wound healing (see Figure 4). Macrophages promote ECM softening in tandem with soluble pro-apoptotic factors, including IL-1B, FGF1, and PGE2, which, in turn, promote myofibroblast apoptosis. However, this is not the only potential fate of myofibroblasts. Myofibroblasts can revert to scar-resolving fibroblasts or temporarily be driven to senescence via CCN family member 1 (CCN1). Myofibroblast apoptosis, reversion to scar-resolving fibroblasts, and conversion to senescent myofibroblasts are all normal fates in the course of normal wound healing. However, myofibroblasts and senescent myofibroblasts can indefinitely escape apoptosis. This pathological turn of events leads to fibrosis and persistent tissue inflammation. Cells in the area are destroyed and replaced with scar tissue. Eventually this leads to tissue and organ failure. Several factors are postulated to account for apoptosis escape associated with the onset of pathological fibrosis. Pro-survival signaling through mechanotransduction pathways and integrin-mediated TGFβ activation have been shown to inhibit IL-1β, FGF1, and prostaglandin E2 (PGE_2_) by shifting the senescence-apoptosis axis toward senescence. Additionally, factors that reinforce matrix stiffening also shift the orientation of myofibroblasts and senescent myofibroblasts toward persistence, survival, and senescence. Stiffening is reinforced by matrix-stabilizing, pro-fibrotic matrix metalloproteinases (MMPs) and crosslink-promoting lysyl oxidases (LOXs). Although MMPs are generally associated with matrix degradation, the reality is more nuanced. Weakening of the ECM by certain MMPs triggers fibroblasts to synthesize and deposit ECM. In addition, these same fibroblasts secrete ECM-crosslinking enzymes such as LOXs [98]. Furthermore, tissue-stabilizing, fibroblast orientation of ECM fibers occurs during normal tissue repair [99], but the disruption of these orienting processes further defines pathological fibrosis. Lastly, advanced glycation products, which form crosslinks between collagen fibrils and also activate the receptor for advanced glycation end products, may play a profound role in the emergence of fibrosis by irreversibly increasing matrix stiffness over time [100].

Myofibroblasts may originate from multiple cell types; however, how they transition from one cell type to another is poorly understood and has not been characterized in the salivary gland. Myofibroblasts can be derived from pericytes, adipocytes, endothelial, epithelial cells, and mesenchymal stem cells (MSCs) [92]. Sun et al. transplanted bone marrow-derived MSCs into injured lung tissue hoping they would differentiate to lung epithelial cells [101]. Instead, these cells differentiated to myofibroblasts, which further exacerbated fibrosis. It was determined that this transition was mitigated by Wnt/β-catenin signaling, which could be attenuated using Dickkopf Wnt signaling pathway inhibitor 1 (DKK1) [102]. Sun’s experimental outcomes speak to the complexities and potential problems investigators may encounter in their efforts to develop therapeutic approaches using MSCs.

However, implantation of MSCs also showed potential to prevent fibrosis. Kim et al. determined adipose-derived MSCs could provoke salivary gland remodeling of fibrotic tissue and other beneficial effects in a mouse model of irradiated salivary gland tissue [103]. Saylam et al. showed adipose-derived MSCs could reduce radiation-induced periductal fibrosis in rats [104]. It is of paramount importance to avoid differentiation of MSCs to myofibroblast after implantation, which may lie in a better understanding of factors that govern myofibroblast conversion in specific contexts.

Engineered salivary gland tissues that incorporate strategies for shifting the senescence/apoptosis axis toward apoptosis can improve the likelihood of successful transplant engraftment and imaginably reverse or halt host tissue fibrosis. Conceptually, this approach to engineering salivary gland tissue is similar in nature to stent technology that not only provides scaffolding architecture to support arterial remodeling, but also integrates strategies for the timed release of chemistries that function to ensure proper engraftment and reduce the risk of complications [105]. Manipulatable factors that disrupt normal shifts of the senescence/apoptosis axis and pathological shifts in salivary gland tissue do exist. Apoptosis, a form of programmed cell death, is regulated by two intertwined pathways: the intrinsic (mitochondrial) and the extrinsic (death receptor) pathways. The BCL-2 protein family plays a pivotal role, particularly in the intrinsic pathway, and can be classified into four categories: sensitizers, pro-survival (anti-apoptotic) proteins, activators, and effectors (pro-apoptotic proteins). Sensitizers inhibit pro-survival proteins that would otherwise inhibit the activators. The activators trigger effectors to initiate mitochondrial outer membrane permeabilization (MOMP) [106] and drive the myofibroblast phenotype toward apoptosis [107]. However, myofibroblasts will not undergo apoptosis in this state if pro-survival proteins are present and sequester pro-apoptotic proteins. BCL-2 homology domain 3 (BH3)-mimetic drugs have been used to circumvent this issue and trigger apoptosis by binding to pro-survival proteins, resulting in apoptosis [108,109].

**Figure 4 bioengineering-11-00028-f004:**
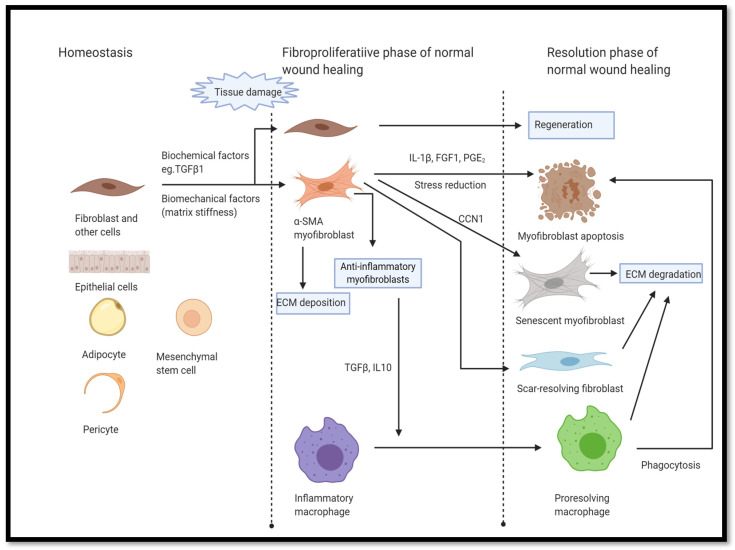
Myofibroblast origins and fate in normal wound healing. Created with Biorender.com [107].

#### 2.2.3. Senolytics and SASP Depressants Support Healing, Reduce Fibrosis, and Improve Transplant Engraftment

Senolytics and SASP depressants have been shown to support healing, reduce fibrosis, and improve transplant engraftment following salivary gland irradiation. Head and neck cancer patients acquire salivary gland damage caused by irradiation. Irradiation boosts mitochondrial ROS production, leading to a reduction in store-operated calcium entry (SOCE), which, in turn, decreases intracellular calcium necessary for the activation of calcium-dependent ion channels driving fluid secretion [110]. When cells are treated with H_2_O_2_, an increase in intracellular Ca^2+^ occurs, which leads to calpain-dependent processing of IL-1α [111], which ultimately leads to the production of inflammatory mediators IL-6 and IL-8 [112], suggesting that reducing oxidative stress would restrict inflammatory gene expression. Chelating Ca^2+^ during senescence inhibits calpain activation and subsequently, IL-1α processing, which attenuates the SASP phenotype. Another study by Ambudkar further validated this concept; H_2_O_2_ treatment resulted in a rapid calcium release from intracellular stores, mediated by the activation of the PLC/IP3/IP3R pathway. Notably, further senescence development was accompanied by persistently elevated [Ca^2+^]*_i_* levels. In H_2_O_2_-treated human MSCs, [Ca^2+^]*_i_* chelation by BAPTA-AM was sufficient to prevent the expansion of the senescence phenotype, decrease endogenous ROS levels, avoid G0/G1 cell-cycle arrest, and finally, retain proliferation [110]. Hai et al. generated a mouse model of irradiation-induced hyposalivation in which a Sonic hedgehog (Shh) gene transfer repressed the irradiation-induced cell-senescence. This effect was attributed to Shh upregulation of DNA repair pathways and decreased oxidative stress [113].

Lastly, numerous studies have demonstrated that pharmacologic (e.g., dasatinib, quercitin) or genetic elimination of senescent cells attenuates fibrosis and function in a variety of organ systems [114,115,116,117,118,119,120,121]. McCarthy et al. found senescence-associated oxidants and calcium drive the secretory phenotype, but antioxidant administration could limit SASP expression by restricting the expression of IL-1α [111]. Alternately, SASP repression strategies may offer a unique benefit, maintaining tumor suppression while eliminating other deleterious SASP-related effects [122,123,124].

## 3. Cell Selection for Salivary Gland Bioengineering

### 3.1. Salivary Gland Cell Lines

Numerous cell lines have been developed for salivary gland basic research studies that inform engineering strategies [125]. These cell lines include human tumor-derived salivary gland cells and rodent immortalized or transformed salivary gland cells (Table 1).

Representative human tumor-derived salivary gland cell lines include HSY, a neoplastic epithelial cell line derived from a thymic mouse tumor after transplantation with surgical specimens of a human parotid gland adenocarcinoma [126], and human salivary gland (HSG), a neoplastic intercalated ductal cell line derived from an irradiated human SMG [127]. Although these human cell lines are useful for biological studies, they are not useful for tissue engineering applications due to their tumor origins. Additionally, caution should be used with the HSG cell line as cross-contamination between this cell line and HeLa cells has been reported [128].

Rodent immortalized or transformed salivary gland cell lines include mouse SIMS [129], mSG-PAC1 and mSG-DUC1 [130], rat submandibular cell lines (SMIE [131] and RSMT-A5 [132]), rat submandibular acinar cell lines (SMG-C6 and SMG-C10 [133]), and rat parotid cell lines (Par-C5 and Par-C10 [134]). Rodent cell lines have been useful for testing cell–cell and cell–material interactions and cell behaviors when seeded on engineered scaffolds for salivary gland regeneration.

**Table 1 bioengineering-11-00028-t001:** Summary of representative salivary gland cell lines.

Species	Cell Lines	Tissue Sources	Cell Types	Characteristics
Human	HSY	Parotid gland adenocarcinoma	Ductal epithelial cells	Form desmosomes and tight junctions (TJs) and exhibit polarization [126,135]; express amylase [136]; respond to muscarinic and β-adrenergic autonomic agonists [137].
	HSG	Irradiated SMG	Intercalated duct epithelial cells	Form desmosomes with sporadic TJ and no AQP expression on plastics [138]; differentiate into acinar structures and express amylase on Matrigel; respond to muscarinic and purinergic agonists; express ductal differentiation markers (EGF, NGF, and renin) [125]; express TJs (claudin-1, -2, -3, -4,ccludingn, JAM-A, and ZO-1) and AQP (AQP5) on Matrigel-coated permeable supports [139]. Report to be contaminated with Hela cells [128].
Mouse	SIMS	A 22-day-old transgenic SMG	Immortalized ductal epithelial cells	Exhibit polarity and express E-cadherin and ZO-1 and duct-specific cytokeratins on Matrigel-coated surfaces; form duct-like structures (cysts) on collagen Type I gels (Col I); When grown on a filter support SIMS cells form a tight monolayer, exhibit vectorial transport function and show exclusive Na+, K(+)-ATPase localization to the basolateral domain [140]; express EGF, NGF, and renin [129].
	SIMP	A 12-day-old PyLT transgenic SMG	Immortalized striated ductal epithelial cells	Exhibit polarity and express E-cadherin and ZO-1 and duct-specific cytokeratins on Matrigel-coated surfaces; form duct-like structures on Col I; express duct-specific cytokeratins and differentiation markers (EGF, NGF, and renin) [140]
	mSG-DUC1	SMG	Genetically modified mice, homozygous for floxed alleles of the integrin α3 subunit	mSG-DUC1 cells express the ductal markers, keratin-7 and keratin-19, and form lumenized spheroids [130].
	mSG-PAC1	SMG	Genetically modified mice, homozygous for floxed alleles of the integrin α3 subunit	Express the ductal markers, keratin-7 and keratin-19, and form lumenized spheroids; express the pro-acinar markers SOX10 and aquaporin-5 [130].
Rat	SMIE	SMG	Immortalized salivary glandular epithelium-like cells	Form TJs on collagen-coated filters [131]; resemble salivary glandular epithelium with an immature lumen; express ZO-1 and E-cadherin, but low level claudin-3 [141]; have a low level transepithelial resistance (TER) that can be regulated by IGF-1 [142]
	RSMT-A5	SMG	Transformed ductal cells	Exhibit a ductal epithelial phenotype and a high density of α1-adrenergic receptors [143].
	SMG-C6	SMG	Immortalized submandibular acinar epithelial cells	Form TJs and desmosomes, enabling polarization [133]; exhibit secretory features (i.e., domes, granules, and canaliculi) and more cytodifferentiation than SMG-C10 [144]; respond to muscarinic and purinergic agonists (but not to α1 agonists) by increasing [Ca^2+^]*_i_* and respond to β-adrenergic agonists by increasing [cAMP]; lack ductal marker cytokeratin 19 expression and exhibit high TER on collagen-coated polycarbonate filters [145].
	SMG-C10	SMG	Immortalized submandibular acinar epithelial cells	Form TJs and desmosomes, enabling polarization; respond to β-adrenergic agonists; exhibit high TER on collagen-coated polycarbonate filters; modulate Na^+^ transport and regulate salivary cell volume [125,133,145,146,147].
	Par-C5	Rat parotid glands	Immortalized acinar epithelial cells	Form layers of plump cells containing intercellular lumen-like invaginations on their medial surfaces; form secretory granules, TJs, intermediate junctions, desmosomes, and microvilli; respond to α1-adrenergic agonists by increasing [cAMP] [133]; respond to cholinergic, muscarinic, and α1-adrenergic agonists by increasing [Ca^2+^]*_i_* [148,149]; express functional amylase [134].
	Par-C10		Immortalized acinar epithelial cells	Form monolayers of cuboidal cells with thick ECM at their bases; form secretory granules, TJs, intermediate junctions, desmosomes, and microvilli [133]; respond to α1-adrenergic agonists by increasing [cAMP] [148]; respond to cholinergic, muscarinic, and α1-adrenergic agonists by increasing [Ca^2+^] [133,148,149]; do not express amylase [134]; exhibit high TER [150,151,152]; express sodium bicarbonate cotransporters and anion exchange proteins on basolateral surfaces, which regulate transepithelial transport. Par-C10 cells achieve transepithelial transport that is sensitive to both intracellular Ca(2+)- and cAMP-dependent stimulation [151]; form 3D differentiated acinar-like spheres on growth-factor-reduced Matrigel, expressing TJs, ion transporters, M3 muscarinic receptors, and AQP3, increasing AQP5 expression under osmotic stress and showing changes in potential difference in response to muscarinic agonist stimulation [152].

A pertinent observation is that most of the cell lines are epithelial. While epithelial cells offer certain utilities, they may not fully capture the complex functionalities of acinar cells in salivary glands. While there is a potential for some of these epithelial cells to dedifferentiate, replicating the natural functions of acinar cells remains a challenge. This inherent limitation underscores the reliance on primary tissues in the field.

The gap in authentic acinar cell cultures impacts salivary gland bioengineering. Acinar cells, being paramount to saliva secretion, are indispensable for reconstituting a functional salivary gland. This gap not only impedes replicating gland function but reinforces dependence on primary tissues.

These challenges provide an argument for exploring induced pluripotent stem cell (iPSC) technology. iPSCs, with their potential to differentiate into diverse cell types, could provide a way for generating acinar-like cells for salivary gland bioengineering, potentially addressing the challenges tied to primary acinar cells or epithelial-derived cell lines.

### 3.2. Primary Salivary Gland Cells

Primary cells have been used to explore cell interactions with scaffolds as well as for implantation in vivo. But critical problems exist with the ability of primary cells to achieve acinar formation in vitro, including a tendency to dedifferentiate when grown on plastic [153]. Salivary epithelial cells also become apoptotic when dissociated into single cell suspensions [154], demanding approaches that will maintain sufficient cell viability. However, single human parotid epithelial cells can differentiate into acinar and ductal structures when grown in a 3D environment [155]. When grown on hyaluronic acid hydrogels, these cells can assemble into acinar lobules, form tight junctions (TJs), develop central lumina, and express α-amylase [156].

A few studies show primary salivary gland cells can be transplanted into living organisms. Human parotid cells grown on polyglycolic acid polymers were implanted subcutaneously into athymic mice [157]. When the polymer scaffolds were retrieved, they had differentiated into acinar structures. Another study transplanted labeled rat SMG into atrophic salivary glands [158]. After several weeks, the labeled SMG cells were detectable over a broad area of the atrophic gland and localized around the acinar and ductal regions, suggesting that salivary gland cells can be transplanted and maintain differentiation. However, in these studies, it was not determined whether the transplanted salivary gland cells were able to function in response to neurotransmitters. In a landmark study, embryonic salivary gland organ “germs” were transplanted in vivo and shown to integrate with existing ductal cells to restore some function [159].

### 3.3. Progenitor Cells of the Developing Salivary Gland

#### 3.3.1. Salivary Gland Stem and Progenitor Cells during Development

The identity of salivary gland stem cells and their capacity to participate in gland development and regenerative responses has been a subject of intense study, since understanding how salivary gland stem and progenitor cells are regulated in development can inform regenerative medicine strategies. Lineage tracing studies, in which cells that are induced to express a fluorescent protein in a specific cell population that can then be traced along with its progeny as development proceeds, have been instrumental in defining the contribution of specific cell populations in developing salivary glands. Cells that express the transcription factors SOX2 and SOX9 and the intermediate filament protein, keratin-5 (K5), prior to emergence of the salivary gland can give rise to all cells in both the SMG and SLG [34,35,42,160,161,162]. During development, cell lineages become more restrictive and K5 is only expressed in ductal progenitor cells and SOX2 and SOX10 in acinar cells and progenitors (proacinar cells). SOX9 plays a pivotal role in determining the cellular differentiation and lineage commitment in the gland development [163].

#### 3.3.2. Stromal-Epithelial Interactions during Development

Salivary gland branching morphogenesis requires interaction of epithelial cells with mesenchyme cells, and mesenchyme cells support the organization and differentiation of salivary gland epithelial cells in organoid cultures [164,165]. Different stromal-derived factors are required to stimulate different aspects of salivary gland development [126]. Although salivary gland progenitors, when combined with Matrigel and EGF to substitute for mesenchyme, could recapitulate branching morphogenesis seen during embryonic development [166], FGF signaling is required for in vivo-like branching and FGF2 signaling in the salivary gland stroma is needed for those cells to support pro-acinar cell differentiation [58,167,168].

### 3.4. Stem Cells for Salivary Gland Tissue Engineering

Studies targeted at increasing regenerative abilities of salivary glands are in progress. In addition, stem cell-based tissue engineering and cell therapy may improve function in damaged salivary glands [169]. Multiple types of stem cells and/or progenitor cells can be considered for regenerative therapies: the unipotent progenitor cells, elusive multipotent adult stem cell including salivary gland stem cells (SGSCs) and MSCs, and induced pluripotent stem cells (iPSCs). There has been an ongoing search for a salivary gland stem cell for many years, based on the assumption that the hematopoietic stem cell-based paradigm is applicable to identify stem cells in all organs. However, recent studies suggest that such multipotent stem cells may not exist in adult salivary glands [170]. MSCs show great potential for tissue regeneration, including salivary gland tissue due to their anti-inflammatory, anti-fibrotic, and regenerative properties, while iPSCs possess the unique capabilities of unlimited self-renewal and the ability to undergo differentiation but with a risk of tumorigenesis. With stem cells, controlling their lineage commitment poses a new set of challenges, which can be better met with tailored biomaterial design strategies that influence transplanted cell fate as well as the host-tissue microenvironment.

#### 3.4.1. Salivary Gland Stem Cells (SGSCs)

Lineage tracing studies in adult glands have enabled the isolation of SGSCs from the ducts of salivary gland tissues. SGSCs are characterized by the expression of a collection of stem cell markers including c-Kit, K5, K14, CD49f, CD90, and CD44 (see Table 2) [171,172]. However, Sui et al. asserted that there is no single, universal, definitive group of SGSC markers [172]. For example, cells located in the striated ducts of the salivary gland expressed stem cell markers identified in other organs, including CD24, CD49f, CD133, and c-Kit [173].

SGSCs can perhaps be best recognized by their behavior, which has been explored in 3D cultures. When human SMG stem/progenitor cell-derived human salispheres were cultured in a collagen I/Matrigel matrix for 2–3 weeks, they formed salivary organoids expressing markers such as cytokeratin, α-amylase, and AQP5 [74]. Human SMG stem/progenitor cells cultured in Matrigel formed aggregates on day 1; when FGF10 was added daily until day 14, they exhibited high expression of gland-specific markers such as AQP5 and Mist1 (acinar markers), α-amylase (functional marker), and α-SMA (salivary myoepithelial marker) and responsiveness to neurotransmitters responsible for salivary secretion [172]. However, the presence of mouse tumor-derived Matrigel makes this system unsuitable for future therapeutic applications. As another option, peptide-modified hyaluronic acid hydrogels supported long-term maintenance of human parotid gland stem/progenitor cells cultured for more than 100 days, showing increased gene expression of acinar markers (e.g., MIST1/BHLHA15, α-amylase/AMY1A) after treatment with β-adrenergic and cholinergic agonists, such as isoproterenol and carbachol for 20 h [171]. However, mature differentiation into secretory acinar cells was not observed.

The potential for implantation of cultured salivary gland organoids was shown by implantation in mice. In vivo transplantation of mouse salivary organoids with E12.5 mouse salivary gland mesenchyme showed natural morphology and saliva secretion, suggesting the role of salivary gland mesenchyme in promoting maturation and function of salivary organoids [172]. Transplantation of mouse submandibular gland tissue-derived progenitor/stem cells were able to regenerate irradiation-damaged salivary gland [173]. Additionally, mouse parotid salivary gland organoid-derived stem cells exhibited irradiation-dose response similar to that of the submandibular gland [174]. Further, Raman spectroscopy has been successfully used to identify the differentiation state of organoids, which may also be generalizable to cells on scaffolds to screen tissue constructs prior to implantation [175].

The therapeutic potential of stem cells is widely recognized. Yet, little is known about the engraftment and capacity of tissues derived from human adult epithelial stem cells. Pringle et al. recently isolated human salivary gland stem/progenitor cells in vitro and observed self-renewal and multilineage differentiation of these cells into organoids. They further demonstrated in vivo functionality, long-term engraftment, and functional restoration of saliva production in irradiated salivary glands in a xenotransplantation model [176]. Additionally, the regenerative potential of stem cells transplanted into irradiated salivary glands was enhanced and further improved by selection for c-Kit expression. This groundbreaking work marks the first instance of salivary gland rescue using salispheres and the first clear demonstration of salivary gland stem/progenitor cell self-renewal and multilineage differentiation into functional organoids.

**Table 2 bioengineering-11-00028-t002:** Salivary gland stem cell markers.

Stem Cell Marker	Salivary Gland Location	Method of Identification	Progenitor or Stem	Reference Number
c-KIT (CD117)	Ducts	Gene expression	Stem	[177]
SCA-1	Ducts	Gene expression	Stem	[177]
Keratin-5 (K5)	Ducts (developing)	Cytoskeletal protein expression, in vivo lineage tracing	Progenitor	[41,178,179]
Keratin-14 (K14)	Ducts (developing)	Cytoskeletal protein expression, in vivo lineage tracing	Progenitor	[179]
LGR5	Ducts (human parotid and submandibular)	Gene expression	Stem	[67]
CD44	Not specified	MSC surface antigen	Stem	[67,180,181,182,183]
CD49f (integrin)	Not specified	MSC surface antigen	Stem	[67,180,181,182,183]
CD90	Not specified	MSC surface antigen	Stem	[67,180,181,182,183]
CD105	Not specified	MSC surface antigen	Stem	[67,180,181,182,183]

#### 3.4.2. Mesenchymal Stem Cells (MSCs)

MSCs are multipotent adult stem cells that can differentiate into several cell types, including adipocytes [184], osteoblasts [185], chondrocytes [186], myocytes [187], cardiomyocytes [188], hepatocytes [189], neuronal cells [190], and salivary gland cells [191], among others. MSCs exhibit anti-inflammatory, anti-fibrotic, and regenerative potential [192,193]. Adult MSCs impact immune T- and B-cell responses through multiple pathways, including T-cell suppression, cytokine regulation, Th1/Th2 balance, Treg regulation, B-cell viability and proliferation, antibody secretion, co-stimulatory molecule production, dendritic cell maturation inhibition, and suppression of IL-2-induced NK cell activation [194]. MSCs that have been tried for salivary gland tissue regeneration include bone marrow-derived MSCs [195], adipose tissue-derived MSCs [196,197,198,199] and salivary gland-derived MSC-like cells [182,200,201]. Denewar et al. demonstrated bone marrow-derived MSCs prevented the development of diabetic-induced hyposalivation in rats [202]. MSCs have several advantages over alternate stem cell types. They are easily obtained from adipose tissues [203] and unlike ESCs and iPSCs, they are not potentially tumorigenic in vivo. However, newer research suggests the origin of MSCs may be an important consideration [204]. Additionally, MSCs can be used for allograft transplantation [205,206,207]. Moreover, by reducing lymphocyte infiltration, fibrotic processes may be mitigated under certain circumstances [194].

Stromal cells derived from the mesenchyme compartment are not only important for development; they also have been used to restore function in hypofunctioning salivary gland. MSCs derived from bone marrow stroma have been shown to improve gland function in a mouse model of Sjögren’s syndrome [208], and adipose-derived MSCs could differentiated into salivary gland acinar-like cells [209] and restored salivary gland function in mice following a radioiodine-induced injury that mimics salivary hypofunction in patients treated for thyroid cancer [103]. In rabbit models, the application of MSCs together with anti-inflammatory agents has been explored with improved function with the dual treatment therapy as an experimental allogenic transplant [210]. Clinical trials are currently underway to evaluate the ability of MSCs to restore salivary function for glands damaged by irradiation [211]. Thus, manipulation of the stromal environment holds promise for clinical applications in salivary hypofunction.

#### 3.4.3. Pluripotent Stem Cells (PSCs): ESCs and iPSCs

ESCs and iPSCs are pluripotent, i.e., they can self-renew and differentiate into somatic cells of all three germ layers—ectoderm, mesoderm, and endoderm [212,213,214,215,216,217,218]. As such, they offer more potency than adult stem cells. Due to ethical concerns and variable immunogenicity associated with ESCs, autologous iPSCs offer a feasible alternative to ESCs [219]. iPSCs were first generated by introducing four transcription factor genes (Oct¾4, SOX2, Klf4, and c-Myc) into mouse and human fibroblasts [216,217]. More advanced methods that produce iPSCs with higher efficiency have since been developed and commercialized [218,220]. The use of both ESCs and iPSCs for more than 14 diseases in clinical trials has shown the promise of PSC-based cell therapies although there are still challenges to overcome, such as heterogeneity and a low, but non-negligible risk of teratoma or carcinoma formation in vivo [221,222]. Teratomas are thought to result from the presence of residual undifferentiated cells and differentiated but still proliferative progenitors, while carcinomas are thought to result from the use of the tumorigenic reprogramming factors (e.g., c-Myc) and viral vectors for generating iPSCs and genetic abnormalities of PSCs. Removing undifferentiated cells and genetically abnormal PSCs prior to implantation and using virus-free gene delivery methods can reduce the likelihood of teratoma development [223,224]. In vitro 3D culture models, including organoids and organ-on-chips, can be utilized to predict tumorigenicity and identify factors to reduce heterogeneity [225,226].

Kawakami et al. co-cultured mouse early ESCs (mEES-6) with mitomycin-treated human salivary gland-derived fibroblasts, attempting to differentiate mEES-6 cells to salivary gland cells (co-SG), and finally, to engraft these co-SG cells into the SMG of immune deficient mice. These co-cultured cells expressed a variety of salivary gland-related markers and could generate new tissues by transplantation in vivo [227]. Additionally, these cells could reconstruct gland architecture in a 3D culture system. Similar results were achieved by co-culturing mouse GFP-iPSCs with E13.5-day SMG cells for 4 days [228]. In a monoculture study, mouse ESCs were differentiated to early salivary gland organoids via stepwise viral induction, focusing on SOX9 and Foxc1, followed by microdissections of protruding buds [229]. The differentiation process was very tedious and inefficient, with loss of salivary organoids/cell aggregates during the frequent medium changes, highlighting the unmet need to differentiate not only mouse but also human iPSCs into mature salivary gland cells efficiently.

## 4. Biomaterials for Salivary Gland Cell Survival, Differentiation, and Engraftment

### 4.1. Cell Support System Overview

The design of an ideal cell support system for salivary gland tissue engineering necessitates a thorough understanding of the native ECM properties and composition. Native decellularized salivary gland tissue has unique properties, whose topography is characterized by honeycomb-like structures with pores in the range of 10–25 μm. Additionally, this tissue presents a notably low indentation modulus, approximately 120 Pa, indicative of its soft, gel-like nature. Such intrinsic properties play a pivotal role in facilitating cellular adhesion, proliferation, and differentiation, ultimately guiding the engineered tissue’s functional outcomes [230,231].

Building on this foundational knowledge, early experiments examined the compatibility of salivary gland cell lines with different polymers and ECM proteins for salivary gland tissue engineering. Aframian et al. performed a 2D experiment wherein HSG cells were cultured on a poly(l-lactic acid) (PLLA), polyglycolide (PGA), or different poly (lactic-co-glycolic acid) (PLGA) co-polymer substrates. The purpose of this study was to examine the growth and morphology of a salivary gland epithelial cell line (HSG) in vitro on several biodegradable substrates as part of an effort to develop an artificial salivary gland. Culture on copolymers alone was unsuccessful, but HSG cells grew particularly well on PLLA coated with ECM proteins, including fibronectin, collagen I, collagen IV, laminin, and gelatin, all of which promoted monolayer growth [232]. Since that time, many different fabrication techniques, such as bioprinting, electrospinning, thermal molding, freeze-drying, and in particular, hydrogel synthesis, have been used to produce scaffolds for salivary gland tissue engineering (Table 3 and Figure 5).

Three dimensional scaffolds that mimic salivary gland tissues can be produced using a variety of top-down or bottom-up approaches. Scaffold fabrication approaches to salivary gland tissue engineering include electrospinning, phase-separation, freeze-drying, hydrogel synthesis, self-assembly, and bioprinting. Top-down approaches typically entail scaffold development, which may integrate essential constituents of native ECM. Cells seeded on such scaffolds are expected to attach, migrate, grow, and proliferate in a systematic way in accord with cues provided by the engineered bioscaffold. On the other hand, bottom-up methods involve the fabrication of tissue building blocks, which can be developed in several ways, cell-encapsulated hydrogels, self-assembled cell aggregates, 2D cell sheets, and bioprinted cells. The specific cell types best suited for transplantation on or within scaffolds are unclear. But immature stem or progenitor cells survive longer than other cell types during the dissociation and transplantation stages than all other tested cell types [249,250]. Based on clinically translatable successes with hematopoietic stem cell transplants [251,252], cell transplantation to reverse incurable disease and regenerate tissue holds promise. However, transplanted cells must survive longer, aggregate less, and integrate into host tissue better than cells have in any study to date [253,254,255], underscoring the unmet need for scaffolds for efficient salivary gland tissue regeneration.

### 4.2. Scaffold Fabrication Approaches to Salivary Gland Tissue Engineering

#### 4.2.1. Electrospinning to Synthesize Fibrous Matrices

Electrospinning is used to fabricate microfibers and nanofibers less than 1000 nm in diameter for the purpose of tissue engineering. A syringe pump, high-voltage source, and a collector plate comprise the basic setup. Generally, a capillary spinning tip is filled with polymer solution with a certain conductivity and viscosity. The solution will stay at the opening of the tip due to surface tension. An electric field is used to oppose the surface tension. Once the force of the electric field exceeds the surface tension forces, a Taylor cone is formed at the opening of the spinning tip, causing a jet of particles to emanate from the tip. Due to molecular cohesion, a continuous stream of liquid results, leading to an unstable and whipping motion of the jet, which is generally termed bending instability. The bending instability and solvent evaporation result in elongation and thinning of fibers as the jet travels to the collector plate. Fiber dimensions can be controlled by regulating solution constituents, humidity, temperature, viscosity, surface tension, and other factors. Numerous synthetic and natural polymers have been used to produce electrospun micro- and nanofibers, e.g., PLLA, PLGA, Poly (glycerol-sebacate) (PGS)/PLGA, polycaprolactone (PCL), alginate, collagen, chitosan, chitin, and silk fibroin [256,257,258,259,260,261,262].

PLGA nanofiber scaffolds were used to investigate physical characteristics of the scaffolds and their influence on cell behavior for application in salivary gland tissue engineering. Sequeira et al. compared PLGA nanofibers from 250 to 1200 nm in diameter [235] and observed a decreased number of focal adhesion complexes in epithelial cells cultured on the nanofibers relative to microfibers or flat surfaces. As the material stiffness was the same for micro- and nanofibers, the topography was assumed to be a significant factor in determining the structure of cell attachments. Additionally, they observed spontaneous self-organization and branching of dissociated embryonic salivary gland cells grown on these nanofibers (Figure 6) [235]. The intent was to increase surface area and better recapitulate the 3D architecture of the basement membrane surrounding spherical acini of salivary gland epithelial cells. They subsequently cultured SIMS ductal and Par-C10 acinar cells in these nanofiber-lined craters, concluding that increasing curvature yielded more polarized cells expressing the tight junction protein, occludin, localized at the apical surface of the cells and increased expression of the water channel protein AQP5 in Par-C10 cells (Figure 7) [234]. Cantara et al. examined effects of PLGA nanofiber scaffolds on cell proliferation and apicobasal polarity. Using murine SIMS ductal and rat SMGC10 parotid acinar salivary gland epithelial cell lines, they observed that cell proliferation was greater on chitosan-coated nanofiber scaffolds than uncoated scaffolds, but that chitosan interfered with apicobasal cell polarity, as indicated by decreased apical localization of the tight junction protein, ZO-1 [263].

Neither salivary gland acinar nor ductal cells fully polarized on these nanofiber scaffolds as indicated by the homogenous membrane distribution of occludin. Functionalization of nanofibers with laminin-111 promoted more mature TJs and demonstrated more apicobasal polarization. To recapitulate the varied functional capabilities of the basement membrane, bifunctional PLGA nanofibers were generated by coating the nanofibers with both chitosan and laminin-111. The signals provided by bifunctional scaffolds prompted a response from both acinar and ductal cell lines, demonstrating the applicability of such scaffolds for epithelial cell types (Figure 8) [263].

#### 4.2.2. Phase Separation to Produce Composite Scaffolds

As scaffolds comprised of one material are rarely sufficient to support and direct cell behavior, the combination of multiple materials is typically needed. Although they have not been widely applied in the development of salivary gland bioscaffolds, phase separation methods can be used to create scaffolds from multiple materials. With these methods, the polymer of choice is dissolved, and then thermal or non-solvent-addition phase separation is induced. In either case, thermodynamic instability results in a two-phase separation. The solvent is generally extracted using water. Next, the temperature is lowered, which causes the polymer-rich phase to solidify into a 3D porous scaffold [264,265].

#### 4.2.3. Freeze-Drying to Fabricate Porous Scaffolds

Freeze-drying, also known as lyophilization, is a method of producing porous scaffolds that does not degrade bioactive molecules [266]. Polymeric and/or proteinaceous materials in solution are frozen, followed by sublimation of the solvent under vacuum, creating materials with a sponge-like structure full of pores. The pore characteristics, including their size, volume fraction, and shape, hinge on several factors. These include the temperature at which the freezing occurs, the concentration of the initial solution, the types of solvent and solute involved, and the way the freezing is managed directionally. Researchers have experimented with a variety of solutions—ranging from water-based to organic, including mixtures with fine particles and solutions where CO_2_ is used in a supercritical fluid state—to produce diverse porous and particulate formations. Innovative techniques such as spray freezing and controlled, directional freezing are pushing the boundaries even further, paving the way for not just porous particles but also materials with pores arranged in a specific alignment, expanding the potential applications of this fascinating process [267].

#### 4.2.4. Hydrogel Synthesis to Form Injectable Cell Delivery Vehicles

Hydrogels are networks of crosslinked hydrophilic polymers that form gels upon hydration. Crosslinked polymers may be chemically synthesized or made from naturally occurring polysaccharides (e.g., alginate, chitosan, and hyaluronic acid) and proteins (e.g., collagen, gelatin, and fibrin). There are many synthetic crosslinking materials including polyvinyl alcohols, polyethylene glycol (PEG), and polyacrylate as well. These materials can be modified in ways that enable transplantation by injection, a cleaner, less invasive means than surgical implantation. To use this method, two critical factors must be considered: first, the mechanical properties of the hydrogel should be optimized for injection, and second, ECM components may be included to prevent apoptosis.

Salivary gland engineers have used CaCl_2_-crosslinked alginate [243,244,268,269], polyacrylamide [242], and PEG hydrogels [241] among others in their research efforts. Miyajima et al. used arginine-glycine-aspartic acid (RGD)-modified alginate hydrogel sheets of different stiffness to demonstrate enhanced bud expansion and cleft formation of submandibular salivary glands. Softer alginate gels facilitated bud expansion and cleft formation, whereas increasing stiffness attenuated these results [243]. RGD-modified alginate hydrogel beads (20, 50, or 100 µm in diameter) also enhanced the ratio of cleft formation and tissue morphogenesis of SMGs [269]. Chitosan-coated alginate hydrogel sheets demonstrated enhanced cell growth, bud expansion, and neural innervation of isolated SMGs [270]. Chitosan also facilitated essential ECM deposition and enhance SMG branch formation [271,272]. Supplemental chitosan added to medium enhanced the morphogenic effects of mesenchyme and mesenchyme-derived growth factors (e.g., FGF7, FGF10, and HGF) on salivary gland epithelial morphogenesis [273] through spatial and temporal regulation of basement membrane [274]. Additionally, egg white-alginate hydrogel discs were synthesized and supported survival and spheroid formation of salivary gland cell lines [275]. Srinivasan et al.’s study showed that parotid cells seeded in HA-based environments self-assembled into acini-like structures expressing functional neurotransmitter receptors [171]. These structures, particularly in 3D hydrogel setups, grew into organized spheroids, exhibiting a range of markers such as CD168/RHAMM, CD44, keratin-5 (K5), keratin-14 (K14), integrin-β1, and α-amylase. Methodologies like staining, immunocytochemistry, and suspension culture techniques were crucial in identifying and verifying the markers and spheroid characteristics.

Hyaluronic acid hydrogels may potentially be used as implantable cell delivery vehicles for salivary gland tissue restoration as well. Pradhan et al. used photo-crosslinked hyaluronic acid hydrogel inserts coupled with perlecan domain IV peptides to culture human parotid gland acinar cells, exhibiting lobular acini-like structures and lumen formation [238].

Shubin et al. employed PEG hydrogels synthesized by step-growth thiol-ene polymerization to encapsulate primary SMG cells in thiol-ene PEG microspheres, which promoted duct and acinar cell proliferation, improved cell viability relative to controls, and maintained the differentiation of salivary gland epithelial cell phenotypes [241]. Fibrin hydrogels have gained traction in the realm of salivary gland bioengineering due to their biomimetic properties and ability to support cellular functions. Fibrin’s inherent ability to guide tissue repair and promote regeneration has been further augmented by chemically conjugating it with laminin-111 peptides [245]. Furthermore, an additional aspect of this research is the potential therapeutic application of these modified fibrin hydrogels in addressing radiation-induced salivary gland damage. Nam et al. assessed the regenerative potential of transdermally injected fibrin solution chemically conjugated with laminin-1 peptides A99 and YIGSR on irradiated salivary glands, forming hydrogels through internal polymerization using endogenous thrombin. The results were promising; the treated irradiated glands showed substantial regeneration, culminating in the restoration of functional salivary tissue. In stark contrast, untreated irradiated glands continued to manifest significant structural and functional degradation [276].

Gelatin-based hydrogels have demonstrated profound potential in the domain of salivary gland regeneration. In a study by Miyaki et al. aimed at creating a model to elucidate the impacts of physiologically active substances on the atrophy and regeneration of salivary gland acinar cells in vivo, acellular gelatin-based hydrogel sheets were employed. These sheets were implanted into resection wounds made in the SMG of Wistar rats. Notably, by day 10 post-implantation, the hydrogel sheets had almost entirely dissipated. The subsequent histochemical examinations revealed that in atrophic regions, a remarkable transition was observed from the initial state of acinar cell atrophy to the emergence of newly formed, mature acinar cells by day 10. These results were further accentuated by the observed transformation of striated and granular ducts into duct-like structures between days 5 and 7. In contrast, necrotic regions showcased a slightly different progression, with a conspicuous destruction of acinar and ductal cells post-resection, followed by the appearance of new acinar cells by day 10 [247]. The significance of this model lies in its capacity to mimic and elucidate the processes of atrophy and subsequent regeneration in the SMG. Moreover, it offers a valuable framework for evaluating the sustained release impacts of physiologically active substances embedded within an implanted hydrogel sheet.

#### 4.2.5. Self-Assembly to Generate Cellular Clusters and Organoids

Scaffold-free methods of developing tissues and organs have emerged, yielding self-assembled and self-organized sets of cells. To implement this approach, cells are selected and exposed over time to a variety of growth regulators. Two distinct forms of scaffold-free technology have evolved, self-assembling processes (SAPs), and self-organizing techniques (SOTs) [277].

In the self-assembling process, non-adherent culture substrates like agarose support high-density seeding, prevent cell-attachment, and encourage cell–cell interaction. Differential adhesion and differential interfacial tension are important concepts in self-assembly, pointing to the idea that cells minimize free energy via cell–cell binding. Consequently, cells with similar surface tension aggregate with one another. Cells with the highest surface tension will sort to the center of a nascent tissue. N-cadherin often plays a role in this process via its expression and localization on cell surfaces. ECM secretions can anchor or free cells from their location. Chemotactic secretions can create concentration gradients that guide cells from one place to another as well [277].

Salivary gland cells have inherent self-assembly properties. This was first revealed when Wei et al. used E13 mouse SMG cells to study tissue assembly. They found that dissociated SMG epithelial cells self-organized into structures that underwent significant branching. Significant insights were garnered by this study. One such insight was that β1-integrin inhibition blocked cell aggregation, but E-cadherin inhibition hampered aggregate compaction. SMG mesenchymal cells added to the epithelial cell cultures facilitated branching and proacinar differentiation [278].

## 5. Potential Engineering Strategies to Improve Salivary Gland Tissue Vascularization, Innervation, and Engraftment

Significant hurdles must be overcome before organ-level tissue engineering becomes clinically translatable. Recapitulating the vascular and neural architecture of engineered salivary glands are big challenges, but new technologies and methods may soon mitigate these two issues. Three dimensional bioprinting, tissue-nanoparticle integration, and mesh electronics are relatively new technologies, which have not been applied to salivary gland engineering in significant measure, but the proof of principle to support their application to salivary gland engineering is abundant. This section will explore these relatively new technologies and prospective approaches for their application to salivary gland engineering.

### 5.1. Prospects for Engineering Vascularized Salivary Gland Tissue

Salivary glands need blood vessels to remain viable as oxygen cannot diffuse more than 200 µm due to mass transfer limitations. Salivary gland thickness surpasses that length along every axis. The human SMGs, for instance, have an anterior-posterior length of 35 ± 5.7 mm, a paramandibular extension to gland depth of 14.3 ± 5.7 mm, and an extension in frontal scanning of 33.7 ± 5.4 mm [279]. Emerging 3D bioprinting technologies can precisely deposit a variety of cell-laden biomaterials with spatiotemporal controls [280]. In another strategy is to produce perfusable vascular tubes, a 3D co-culture of vasculogenic cells (e.g., human umbilical vascular endothelial cells, HUVECs) and human adipose-derived stem cells (hASCs) within a synthetically modified fibrin hydrogel was developed [281].

### 5.2. Prospects for Engineering Innervated Salivary Gland Tissue

Innervation of engineered tissue has been a long-standing, but an elusive goal of tissue engineers. As both salivary gland development and saliva secretion depend upon sympathetic and parasympathetic innervation, engineered salivary glands must have the capacity to interface with the nervous system. Evidence indicates that normal salivary gland development will not occur in the absence of the PSG [41,282,283].

#### 5.2.1. Mesh Electronics and Bio-Hybrid Systems 

Normal development and maintenance of salivary gland tissue is severely undermined when the parasympathetic ganglion is damaged [38,284]. Mesh electronics may offer a way to circumvent salivary gland innervation problems that emerge after salivary gland damage. Mesh electronics are a class of ultra-flexible and scalable neural probes originally designed to seamlessly integrate with the neural tissues (Figure 9). These semiconductor electronics resemble a 3D mesh or network, allowing them to be implanted in the brain (or other tissues) with minimal inflammation or damage over long durations. Their design aims to achieve a biologically compatible interface, where the implanted electronic mesh can move with the tissue, reducing the relative motion between the device and the surrounding neurons. This ensures long-term stability and a reduced immune response. Bridging is enabled by the structural and mechanical properties of mesh electronics, which mimic native neurons. Mesh electronics are small enough to be injected directly into tissue or integrated into engineered tissues and transplanted. These probes are neuro-attractive and non-immunogenic. Further, they are capable of long-term mapping and modulation of neural activity [285,286]. Mesh electronics show promise for restoration of function in damaged glands and for creating functional engineered glands.

#### 5.2.2. Biocompatible, Endocytosed Nanotubes in Salivary Gland Tissue

Nanotubes can be comprised of a single sheet of atoms or multiple layers formed into a tube having a diameter in the nanometer range. Goldman et al. discovered that multiwall inorganic tungsten disulfide ([WS2], INT-WS2) nanotubes, which are 40–150 nm in diameter/75–100 nm in length and fullerene-like nanoparticles (IF-WS2), which are 120–150 nm in diameter were biocompatible with rat submandibular cells. Transmission electron microscopy (TEM) images indicated nanoparticle (NP) endocytosis and accumulation in cytoplasmic vesicles, suggesting promising future uses as these NPs have many potential medical applications [287]. NPs of different compounds form nanotubes and fullerene-like particles and can be functionalized with proteins and other biomolecules, enabling targeted drug delivery and bioimaging capabilities [288]. These nanomaterials have superior mechanical and tribological (interacting surface in relative motion) properties. Additionally, carbon nanotubes have demonstrated the potential to induce axonal regeneration and peripheral nerve repair because of their unique properties, such as biocompatibility, electrical conductivity, and flexibility [289].

#### 5.2.3. Bioprinting of Neurons and Innervation of Tissues

Bioprinting offers the possibility of printing stem cells, nerve cells, and nerve conduits for integration with engineered salivary gland tissue. Owens et al. developed a synthetic nerve graft using murine bone-marrow stem cells (BMSCs) and Schwann cells. The cells were cast into 500 µm diameter tubes then loaded into an extrusion bioprinter, which formed Schwann cell tubes surrounded by BMSCs [283]. Lorber et al. printed rat ganglion cells and glia using an inkjet printing system [290]. Pateman et al. used micro-stereolithography to print PEG-based nerve guidance conduits for nerve repair studies, which performed similarly to autograft controls [291]. As bioprinting technology continues to improve, we can anticipate more papers and technological advances in this important area to facilitate innervation of engineered tissues.

#### 5.2.4. Nanoparticles to Offer Essential Spatiotemporal Control over Scaffold Development and Engraftment

NPs range in size from 1 to 100 nm, creating extremely high surface area to volume ratios. The surface area and size of NPs make them highly mobile and interactive, with the potential for highly tunable interactions. The NP surface and its interior can be functionalized for many tissue engineering-relevant purposes including cell-specific targeting and penetration, heat production, vibration, antimicrobial activity, contrast, magnetic control, and conductance. These particles can be designed with biocompatibility and non-immunogenicity in mind and can be engineered to mimic the size of ECM components. Researchers have developed NP applications specific to tissue engineering fields across several basic areas, including biological, electrical, and mechanical-property enhancement, mechanotransduction stimulators, gene delivery, magnetic cell patterning, 3D tissue construction, and biomolecular detection. NPs can be directed and delivered to salivary gland tissue or integrated into engineered salivary gland tissue and are demonstrably biocompatible and easily endocytosed [292].

NPs can improve the engraftment environment by reducing apoptosis and can also be used to promote PSG innervation through the controlled release of growth factors to drive branching morphogenesis of neurites. Further, Arany et al. targeted the pro-apoptotic PKCδ gene using a novel, pH-responsive nanoparticle complexed with siRNA. The knockdown of PKCδ not only reduced the number of apoptotic cells during the acute phase of radiation damage, but also markedly improved saliva secretion at 3 months in irradiated animals. Treatment was administered prior to ionizing radiation [293]. Varghese et al. developed a protocol for nanoparticle delivery to salivary gland tissue using retroductal injection of the SMG via Wharton’s duct and parotid gland via Stetson’s duct. As nanoparticles can be delivered in combinations and with spatiotemporal controls, future advancements will further increase their applicability for salivary gland tissue engineering and other medical applications [292].

#### 5.2.5. Injectable Engineered Salivary Gland Transplants

Injectable scaffolds allow for minimally invasive cell delivery in vivo and can be used to increase survival rates upon cell implantation. However, this process can be hampered by mechanical stress during cell injection [294,295]. Shear-thinning biomaterials are materials that decrease in viscosity or become less viscous when subjected to increasing shear stress. This is a non-Newtonian flow property wherein the material becomes more fluid-like under mechanical stress or agitation and returns to its more viscous or semi-solid state once the stress is removed. Shear-thinning biomaterials, such as hyaluronic acid [296] and alginate [297,298], have been shown to form a lubricating layer on the syringe wall, which reduces resistance to flow, which, in turn, leads to increased cell viability after injection [298]. Once these cells are transferred, they are still subject to apoptosis as many cells undergo integrin-mediated apoptosis, or anoikis, in the absence of ECM binding [299,300]. Injectant inclusion of ECM components, such as collagen, laminin, fibronectin, and HA, can prevent this anoikis. Other compounds can also improve post-transplant survival, and biomaterials can be modified to mimic ECM-mediated signaling through inclusion of integrin-binding domains. RGD is a well-known integrin-binding domain that binds to at least eight different integrin complexes [301,302]. These integrin-binding domains offer many benefits. First, RGD is the main integrin-binding domain present within ECM proteins such as fibronectin and vitronectin; hence, it is widely recognized by many cell types. Second, short peptides like RGD are more stable than their corresponding proteins. Third, as functionalization components, they easily fit biomaterial spatial and conformational parameters [303,304]. Studies have shown a marked increase in in vitro MSC survival following the use of RGD-modified alginate [298]. Like RGD, the laminin-derived peptides, isoleucine-lysine-valine-alanine-valine (IKVAV) and tyrosine-isoleucine-glycine-serine-arginine (YIGSR), and the collagen-derived peptide glycine-phenylalanine-hydroxyproline-glycine-glutamic acid-arginine (GFOGER) can improve cell survival for specific cell types [303]. Two examples are laminin- and collagen-mimetic peptides, which can be used for neural cell transplantation and musculoskeletal system transplantation, respectively [305,306]. Injectable scaffolds hold promise for salivary gland engineering and also for repair of damaged glands.

An exploration of the various strategies aimed at enhancing glandular vascularization, innervation, and engraftment would not be complete without consideration of how these methods would be applied in different pathological environments, particularly radiation-damaged versus diseased or senescent ones. Radiation damage often results in acute and chronic inflammation, vascular damage, and fibrosis [307]. Strategies employed for this environment need to not only promote tissue regeneration but also combat the detrimental effects of radiation. For instance, antioxidants, anti-inflammatory agents, and growth factors might be incorporated into the tissue engineering approach to mitigate the radiation-induced damage. Furthermore, given the compromised vasculature post-radiation, strategies that promote early rapid vascularization will be vital to ensure the viability of the engineered tissue in the radiation-damaged environment, or a diseased, or senescent glandular environment that is characterized by cell aging, reduced proliferative capacity, and often, chronic inflammation [69]. In such a context, strategies should focus on rejuvenating the tissue, possibly through the introduction of younger, more proliferative cells or stem cells. Additionally, given the chronic inflammation often seen in these settings, anti-inflammatory approaches will also be of value. Tissue engineering solutions here might prioritize restoring normal cellular function, potentially utilizing signaling molecules that combat senescence or promote cellular rejuvenation. Lastly, an emerging class of compounds known as senolytics offer promise in reducing the effects of senescent cell burden in diseased or senescent tissue [308]. In summary, while the goal remains the restoration of gland function, the specific challenges posed by radiation damage versus inherent glandular disease or senescence necessitate tailored strategies. The effectiveness of any tissue engineering approach will hinge on its adaptability to the unique challenges of the environment under consideration.

**Figure 9 bioengineering-11-00028-f009:**
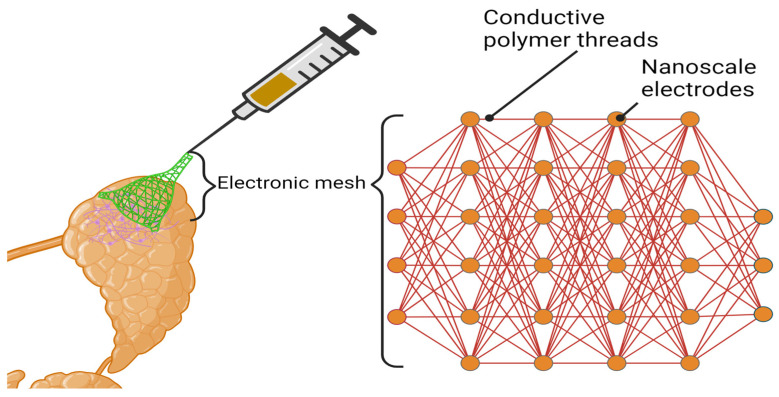
Electronic mesh schematic (created using Biorender, Biorender.com) [309].

## 6. Conclusions and Future Perspectives

As current treatments for salivary hypofunction are inadequate and only offer transient relief, regenerative medicine-based treatments are being developed. Efforts to date have included gene therapy, stem/progenitor cell-based therapy, and tissue engineering strategies. Understanding salivary gland development and its relevance to normal and dysregulated wound healing provides a foundation for the development of engineered salivary gland tissues. Salivary gland, stromal cells, nerves, and vasculature engage in reciprocal signaling that leads to branching morphogenesis of the epithelium, integration with the vascular and nervous systems, and the eventual elaboration of mature secretory salivary glands. Depending on the cause, hypofunctioning salivary gland tissue can be characterized histologically by degenerated acinar epithelial cells, an increase in the ductal/acinar epithelial ratio, exaggerated numbers of senescent cells, fibrotic stromal tissue, and increased immune infiltration. Efforts to develop salivary gland tissue models must carefully consider what biomaterials and fabrication methods will be used to provide a substrate for culturing cells, and many natural, synthetic, and semi-synthetic materials have been used to create scaffolds for elucidation of cell interactions with scaffolds and for implantation in vivo. Fabrication strategies have included electrospinning, phase-separation, freeze-drying, self-assembly, enhanced hydrogels, photolithography, and bioprinting. For eventual clinical application, bioengineering a tissue greater than 200 µm thick that is perfusable and innervated remains a major challenge in the development of engineered salivary gland tissue that fully recapitulates the features inherent in the natural tissue. Significant progress has been made in overcoming these hurdles using innovative technologies that include bioprinting, microfluidics, the use of nanotubes and NPs, mesh electronics, and other nascent technologies (see Table 4).

## Figures and Tables

**Figure 1 bioengineering-11-00028-f001:**
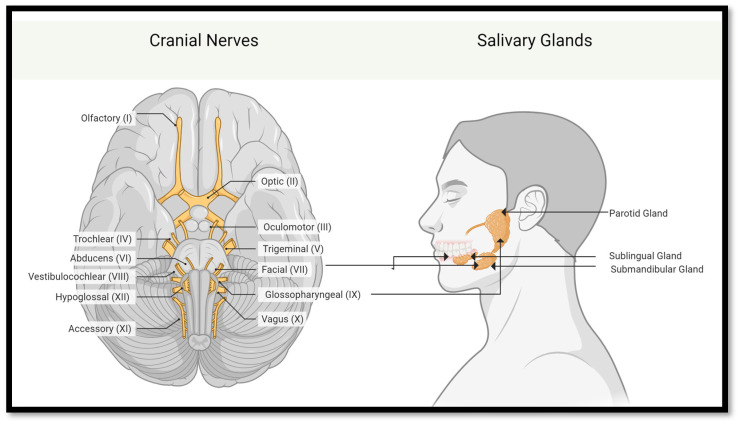
Primary salivary glands. There are three pairs of major salivary glands—parotid, submandibular, and sublingual—innervated by the facial and glossopharyngeal nerves. Created with BioRender.com (2023).

**Figure 2 bioengineering-11-00028-f002:**
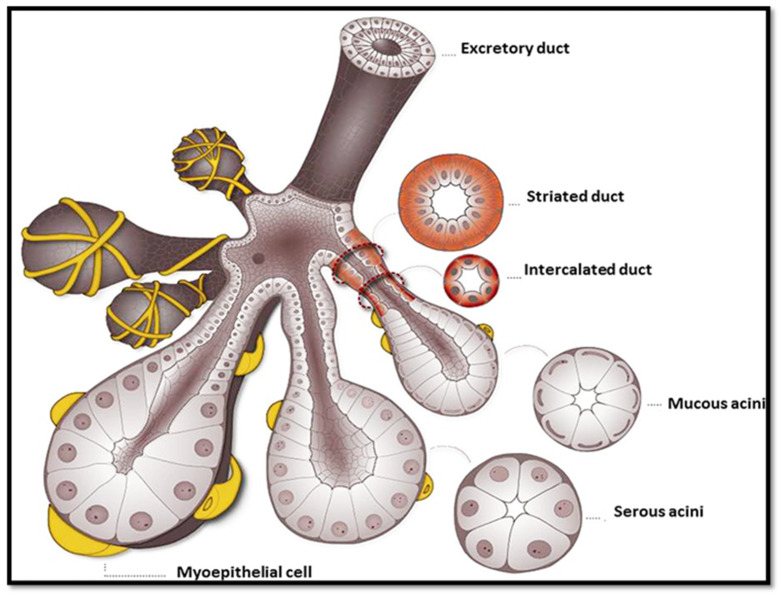
The adenomere functional unit of the human salivary gland. Reprinted with permission from de Paula et al., 2017. © 2017 Wiley Periodicals [13].

**Figure 3 bioengineering-11-00028-f003:**
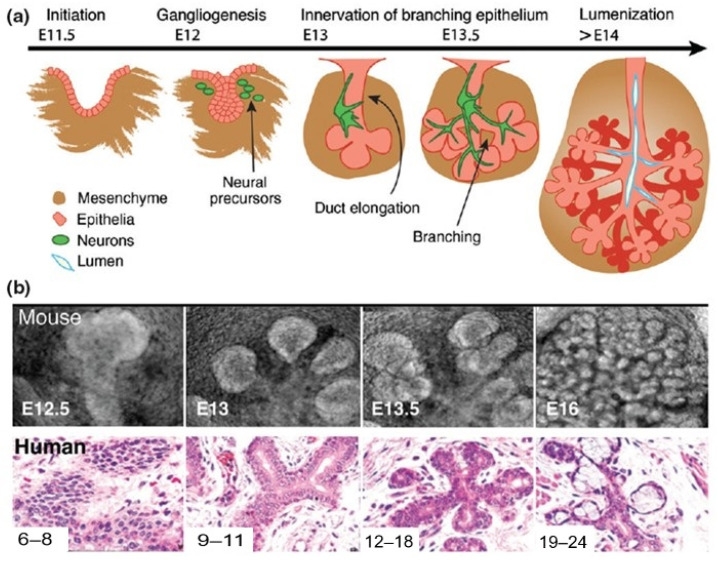
Mouse and human salivary glands develop through the process of branching morphogenesis. (**a**) Schematic stages of mouse salivary gland development. (**b**) Histological representations of mouse and human saliavry gland development. (Top panel) Mouse development measured in days. E = embryonic day. (Bottom panel) Human development measured in weeks. Reprinted with permission from Mattingly et al., 2015. © 2015 Wiley Periodicals [22].

**Figure 5 bioengineering-11-00028-f005:**
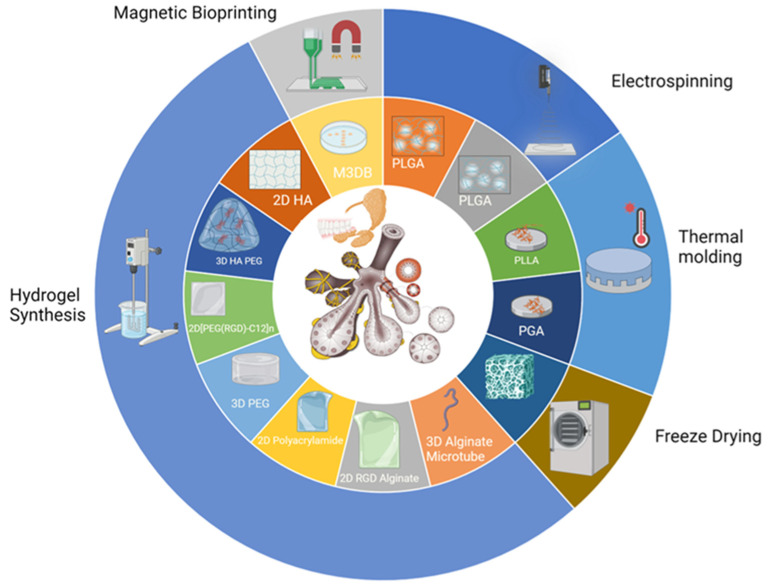
Major approaches to biofabrication of salivary gland tissues (Created using Biorender, Biorender.com).

**Figure 6 bioengineering-11-00028-f006:**
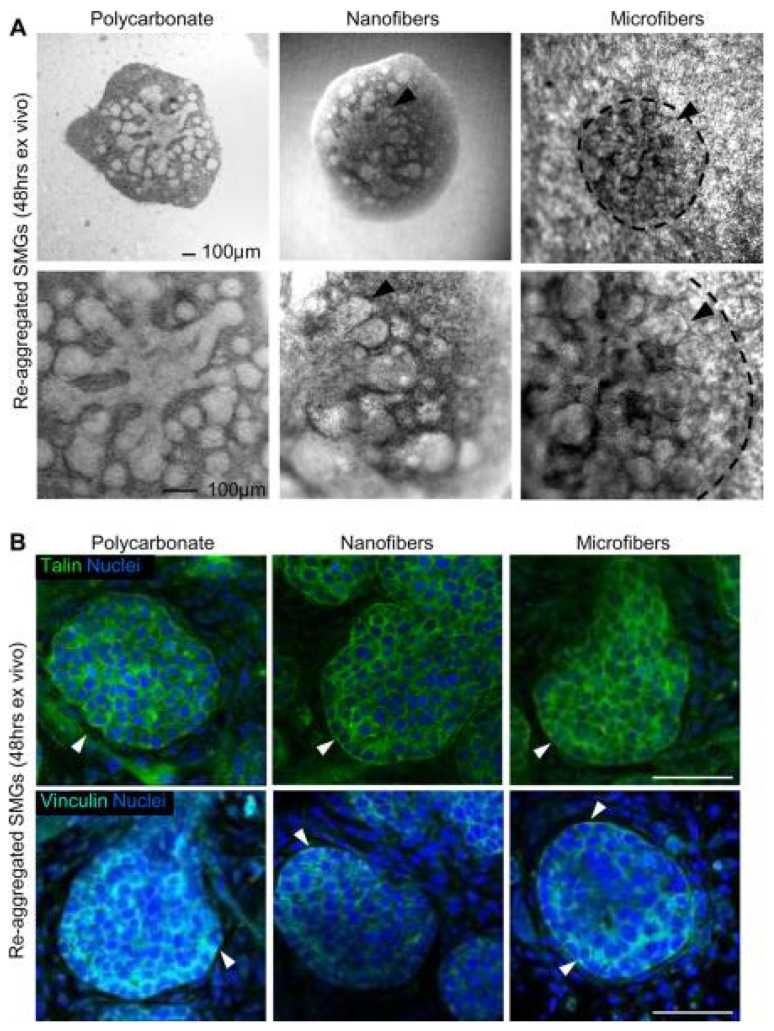
Nanofiber scaffolds promote self-organization and branching morphogenesis of dissociated embryonic salivary gland cells. (**A**) Bright field images of spontaneously re-aggregated cell pellets from dissociated embryonic day 13 (E13) salivary gland cells cultured on polycarbonate filter membrane, PLGA nanofibers, and microfibers for 48 h. Dashed lines outline the morphology of re-aggregated structures on microfibers to compensate for the interference with light microscopy. (**B**) Confocal images through the equatorial section of re-aggregated buds immunostaining for talin (green, **top panels**) or vinculin (cyan, **bottom panels**), co-stained for nuclei (DAPI, blue), showed diffuse cortical expression with stronger staining along the basal cell membranes at the bud periphery (arrowheads), similar to that observed in intact glands, scale bars = 50 μm. Reprinted with permission from Sequeira et al., 2012. © 2012 Elsevier [235].

**Figure 7 bioengineering-11-00028-f007:**
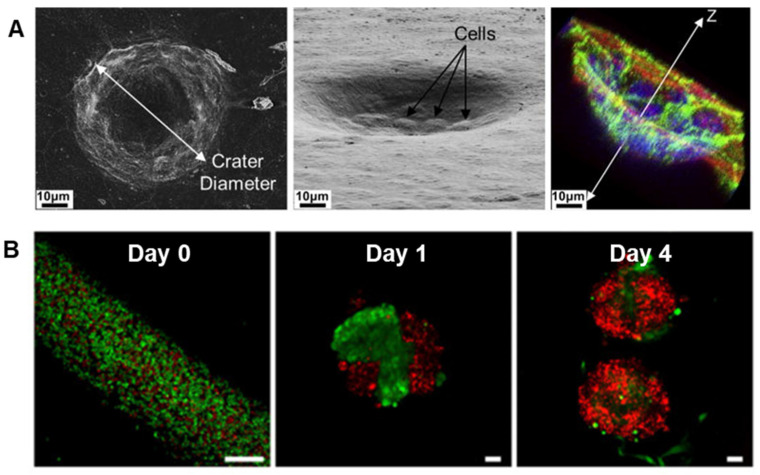
Biomaterials can control shape of organized tissue constructs. (**A**) Confocal image (right) of SIMS cells stained for F-actin (green, phalloidin) and nuclei (blue, DAPI) grown in 30 μm nanofibrous craters (SEM of top view, left and angled view, middle) for 96 h, scale bar = 10 μm. The Z-plane is denoted by the arrow. Reprinted with permission from Soscia et al., 2013. © 2022 Elsevier [234]. (**B**) Confocal images of CellTracker™ Red CMTPX-labeled SIMS cells and CellTracker™ Green CMFDA-labeled NIH 3T3 fibroblasts showed cellular organization after co-cultured in microtubes for 4 days, scale bar = 250 µm. Reprinted and modified with permission from Jorgensen et al., 2022. © 2022 by the authors [244].

**Figure 8 bioengineering-11-00028-f008:**
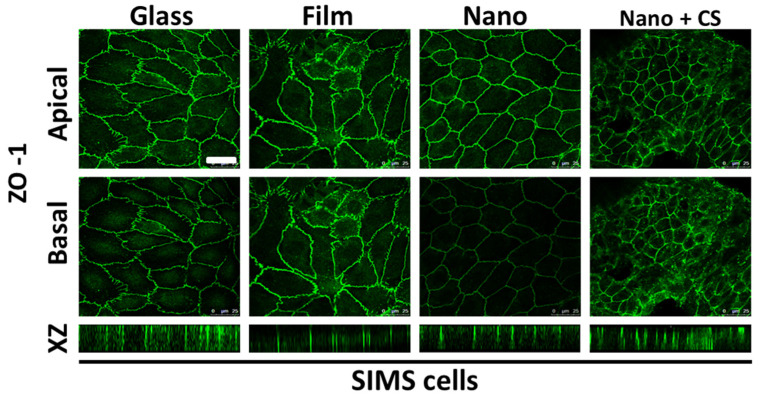
Cell behavior in organoids can be achieved through cell surface modifications of nanofibrous scaffolds. Confocal images of immunostained SIMS cells seeded on PLGA nanofibers (Nano) showed apical restriction of ZO-1 (green), which was disrupted by chitosan-modified nanofibers (Nano + CS) compared to 2D culture on glass or PLGA film. Scale bar = 25 µm. Reprinted with permission from Cantara et al., 2012. © 2012 Elsevier [263].

**Table 3 bioengineering-11-00028-t003:** Major approaches to biofabrication of scaffolds for salivary gland tissue engineering.

Fabrication Method	Biomaterial and Dimensionality	Cell Type	Advantages and Disadvantages
Thermal molding	2D PLLA and PGA (flat disks)	HSG	Pros: Biodegradable; able to form into coverslip-like disks suitable for cell seeding; versatile permitting melt-processing polymer pellets between sheets of aluminum foil using a Carver press at 350 °F, 450 °F, 175 °F, and 200 °F for PLLA, PGA, 50/50 PLGA, and 85/15 PLGA, respectively [232,233]. Cons: Require ECM coating to support cell attachment; lack 3D cues.
Electrospinning	PLGA (fibrous scaffolds)	SIMS	Pros: Provide topographic cues (e.g., nanofibers, curvature); exhibit more rounded and clustered cell shape vs. 2D flat disks; enhance cell-polarization effects and expression of water channel proteins [234,235]. Cons: Require laminin coating for cell polarization and tight junctions; lack in vivo-like viscoelasticity.
	PLGA (fibrous scaffolds)	Par-C10	
Freeze-drying	Silk fibroin (porous scaffolds)	Primary salivary gland epithelial cells from rat SMG and parotid gland	Pros: Provide topographical cues; promote epithelial cell growth; facilitate the secretion of ECM proteins; retain the differentiated function [236,237]. Cons: Require fibronectin coating; mimic the basement membrane for epithelial cells but might not be ideal for stromal cell culture; lack in vivo-like viscoelasticity.
Hydrogel synthesis	HA hydrogels (cell culture insert)	Primary human salivary gland acinar-like cells from the parotid gland	Pros: Mimic the hydrogel component of ECM; exhibit acini-like structures with tight junctions, α-amylase expression, and an apoptotic central lumen on HA gels with an elastic modulus of 2000 Pa and incorporating peptide derived from domain IV of perlecan [238]. Cons: Require coupling bioactive peptides; or form acinar-like structures that are less organized or slowly growing in 2D or 2.5D than those in 3D HA hydrogels.
	3D HA/PEG hydrogels (cell culture insert)	Primary human salivary gland acinar-like cells from the parotid gland	Pros: Provide 3D microenvironment for encapsulated cells; facilitate cell self-assembly into acini-like spheroids of ~50 µm in size; demonstrate neurotransmitter-stimulated protein secretion and fluid production; integration in an in vivo rat model with no obvious signs of inflammation [239,240]. Cons: Indicate reverse polarity; lack essential machinery for full salivary restoration.
	[PEG(RGD)-C12]_n_ microfibers	Human primary salivary gland myoepithelial cells	Pros: Fabricate meter-long multiblock copolymer microfibers via straightforward interfacial bioorthogonal polymerization; provide guidance cues for the attachment and elongation of myoepithelial cells [17]. Cons: Cannot use for cell encapsulation; culture cells on the surface two dimensionally rather inside the microfiber three dimensionally.
	3D PEG hydrogels	A mixture of primary acinar and ductal cells from mouse SMG	Pros: Improve cell viability and proliferation and facilitate cell–cell contacts by encapsulation of pre-assembled spheroids [241]. Cons: Remain as single cells without forming organized acini-like structures after cell encapsulation in 3D PEG hydrogels.
	2D polyacrylamide gels	Mouse E13 SMG	Pros: Promote branching morphogenesis; partially rescue acini structure and differentiation by transferring glands from stiff to soft gels or by adding exogenous TGFβ1 [242] Cons: Require addition of exogenous TGFβ1 to polyacrylamide gels for partial acini structure rescue; lack 3D cues.
	2D RGD-modified alginate hydrogel sheet	Mouse E13 mesenchymal cells and SMG	Pros: Promote mesenchymal (not epithelial) cell adhesion by RGD surface modification; enhance the bud expansion and cleft formation in SMG by softer gels, whereas stiffer gels attenuate them and decrease gene expression of FGF7 and FGF10; partially rescue acini structure and differentiation by adding exogenous FGF7 or FGF10, or by transferring SMGs from stiff to soft gels [243]. Cons: Stiffer RGD-modified alginate hydrogel sheets attenuate bud expansion and decrease gene expression of FGF7 and FGF10.
	3D Alginate hydrogel microtubes	Co-culture of SIMS with mouse NIH 3T3 fibroblasts or E16 mesenchyme cells	Pros: Provide 3D microenvironment in hydrogel that is easy to handle; allow for high density cell growth; facilitate 3D mesenchymal-epithelial interaction; allow salivary gland epithelial cell organization into 3D cavitated structures with lumen formation; exhibit potential for formation of uniform organoids and functional units [244]. Cons: Require 3D arrangement of microtubes with organoids and additional elements to construct the full machinery of the salivary gland.
	2D Fibrin-based hydrogels	Par-C10	Pros: Support differentiation of salivary gland cell clusters with mature lumens [245] Cons: 2D culture on the hydrogel surface; require laminin-111 peptide-modification.
	Fibrin-based hydrogels	Acellular, laminin I peptide functionalized	Pros: Mitigate the risk of tumor development; results in restoration of functional salivary tissue [246]. Cons: Require decoration with laminin-1 peptides; require injection of liquid followed by internal gelation to avoid hydrogel clogging the needle.
	Gelatin-based hydrogel sheet	Acellular, controlled release of growth factors (EGF, FGF, NGF)	Pros: Demonstrate atrophy and regeneration of the SMG; enable observation of effects of sustained release of physiologically active substances contained within an implanted hydrogel sheet. Cons: Collapse of the hydrogel mesh began by day 7, in conjunction with invasion of surrounding fibrotic connective tissue, without regeneration of the salivary gland tissue [247].
Cryoelectrospinning	3D Alginate-elastin cryoelectrospinning scaffolds	NIH 3T3 fibroblasts	Pros: Produce porous nanofiber-sponge scaffolds that recapitulate the topography and viscoelasticity of salivary gland ECM; allow cell penetration deeply and 3D culture; support stromal cell viability and homeostatic marker expression [230]. Cons: Require dynamic seeding to achieve high seeding efficiency; require dynamic culture to achieve high density cell growth.
Bioprinting	Magnetic 3D bioprinting (M3DB)	Neural crest-derived MSCs and human dental pulp stem cells	Pros: Develop innervated secretory epithelial organoids in the presence of FGF using cells tagged with magnetic nanoparticles that are ordered using magnetic dots. Cons: Require magnetic nanoparticles; challenge to determine an apicobasal polarization due to tightly packed epithelial cells; exhibit limited vascularization in the organoids [248].

**Table 4 bioengineering-11-00028-t004:** Future perspectives of salivary gland bioengineering.

Category	Details and Strategies
Origins and development	Ectodermal origin of major salivary glands Role of epithelial, stromal, endothelial, and nerve cells
Histological features of hypofunctioning tissue	Degenerated acinar epithelial cells; increased ductal/acinar ratio Fibrotic stromal tissue Increased immune infiltration
Fabrication strategies for scaffolds	Electrospinning, phase separation, and freeze-drying Self-assembly, enhanced hydrogels, and photolithography Bioprinting
Major challenges	Creating tissue > 200 µm thick that is perfusable and innervated Integration with the in vivo environment Integration with vasculature and nervous systems
Emerging technologies	Bioprinting, microfluidics, and cryoelectrospinning Nanotubes, mesh electronics
Outlook	Redefining current techniques Precise integration of cellular components Exploring new approaches on the horizon in regenerative medicine for salivary hypofunction

## Data Availability

Not applicable.

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
