# Peer review of "Salivary Gland Bioengineering"

_bioengineering, 2023, doi:10.3390/bioengineering11010028_

Round 1

Reviewer 1 Report

Comments and Suggestions for Authors

This   is a well-written and organized, comprehensive review. It can be published after the authors consider the following minor concerns.

The are numerous places where a space could be added before a new section, e.g. sec 2.1.2

The document is text heavy – some more pictures would be appreciated.

Line   40 – “poor oral health contributes to….”,   is it a cause or associated with?

Line   80 something – adenomere may be less familiar than an acinus or acini, should it be more specifically defined?

Lines 93-94 – the statement is a bit confusing.  This could be helped by a following statement indicating  which glands produce more mucins   vs. watery fluid.

Fig.2 although it is   a reproduced figure it could be adapted to define the nerves and myoepithelial cells.    Also   the figure suggests that individual acini are either serous or   mucous, is that true?  If not, perhaps that should be noted in the text or caption

Line 108 – how does pumping of Na+ and Cl- make saliva hypotonic?  Are HCO3- and K+ are pumped into saliva at a lower mole basis than Na+ and Cl- are pumped out?

Fig.2  caption De Paula, should be de Paula based on ref 13

Line 149- the saliva reduction should have a ref.

Line 189 – “B” should be FGFR2b

Line 200 – human or mouse adult…

Line 218  - stem cell pools – where do these exist in the gland? There is a lot of discussion about stem cells and the authors even appreciate the debate as to whether they exist in the gland or not.  I wonder if a more focused section on stem cells in the SG should precede the all the literature cited with regard to their use.  There is discussion of iPSC and ESCs later on with respect to other tissues regeneration. I don’t believe protocols exist yet to differential SG from these cells.

Fig 4, I may have missed it but is it reference to Fig. 4 in the text?

Line 348 – the  Sun paper used MSCs to …. What was the source of these cells since it makes a difference.

Lines 373 to 383 about apoptosis seems a bit out of place – not sure where this discussion fits better.

Line 379 – needs a space before [79].

Sec 3.1 – I think it should be noted that most all of these “cell lines” are epithelial cells. I suppose it is possible for them to dedifferentiate into an acinar cells but this is a stretch in my mind. It is the lack of acinar cell culture that hinders the field and drives this use of primary tissues. Perhaps some comment regarding this that motivates development of IPSC technology is needed.

Line 477   - typo, extra space

Section 3.4.1 maybe there needs to be   a separate section that defines accepted markers of acinar,  ductal and other cell types???

Line 542 there should be a refs for the various cell types that MSC can differentiate to.

Line 661   ref 211 is incomplete

Lines 671- 693 – be nice if the authors could add a table that summarizes important material properties of the salivary gland, stiffness etc.  Vague statements are made about biomaterial stiffness.

Section 5 is hugely important for all TE application although the discussion here is rather modest.  This is a topic for a specific review.  What is presented is worth keeping   - be nice if the text could include some pictures. 

Line 905 – can you define what mesh electronics are.  I have an idea but maybe a picture or note what they are made of?

Line 961 – can you define shear-thinning biomaterials.

Reviewer 2 Report

Comments and Suggestions for Authors

Comments for Review of the Manuscript, “Salivary Gland Bioengineering”

Overall, this is a nice paper touch on multiple topics without going into details, and the ones that they do go into detail they have not highlighted the advantages or disadvantages of the strategies of glandular regeneration. Overall, the manuscript needs to be revised before publication.

Specific Comments:

P1L44: What kind of side effects occur after administering these treatments? Are they severe enough that patients would be willing to submit to implanted salivary glands?

P2L70: Can you provide a diagram showing the difference between the innervation of the salivary glands?

While you provide tables listing the difference between cell types and scaffolds, and this is done quite well, it would be beneficial to include a compound figure for the references or to make a composite schematic showing the difference in results.  A composite figure would help visualize the gland reconstruction and development, and schematics would show a visual for the differences in scaffolding.

Additionally, please give some kind of table or visual when discussing future prospects of salivary gland engineering.

P3L112 Branching morphogenesis should be discussed in more detail. Especially the molecular mechanisms that drive morphogenesis. For example, the SOX2/9/10 transcription factors that are involved in branching morphogenesis.

P4L124 This section describes salivary development through the different cell types. Yet there is no mention of the effect of endothelial cells even though it is in the title of the section.

P4L134 Would be useful to know if there are any markers that define the mesenchymal condensate and the thickening of the oral epithelium lineages as they are important for the initiation of the bud.

P6L204 The authors give a good accounting of fibrosis and senescence; however, it would be important to know if cellular senescence affects a certain cell type namely either the acinar/ductal/myoepithelial/ or basal cell populations in particular.

P12L456 This section should also include studies involving SOX9 progenitors as a parent population to all gland cells.

P15L595 ‘stepwise viral induction followed by microdissections of protruding buds’ may be a better way to put it to emphasize the inefficiency of the protocol to generate the organoids.

P15L600 This section needs more diverse hydrogel work to be incorporated into the manuscript. As of now the authors have only referred to synthetic scaffolds or HA/alginate scaffolds, this section should also include work on fibrin hydrogels and gelatin-based hydrogels. It would also be more useful to restructure Table 2 observation section into an advantages/disadvantages section to highlight the strengths and weaknesses of the approaches.

P20L833 The authors have given a good account on strategies to improve glandular vascularization, innervation and engraftment. It is also imperative to discuss how these strategies would work in a radiation damaged v/s diseased/senescent environment.

Comments on the Quality of English Language

NA

Reviewer 3 Report

Comments and Suggestions for Authors

. P1. Page 4 shows that when the mouse embryo is 11 days old and the human embryo is 30 days old, the nerve ridge cells receive a signal to make mesenchymal condensate in the oral epithelium. Humans and mice have different species, so of course the mechanisms by which various signals work are different. I think it would be helpful to understand if the authors write down in detail what kind of signal the response to form a mesenchymal condensate is activated.

2.    Page 19 shows that spheroids that secrete protein and fluid in vitro were formed and transplanted into mice, and these spheroids expressed spheroid structure and marker for a week after in vivo transplantation into the glands under the ears of mice. It would be better to mention on what basis the marker of spheroid was expressed, or whether there is a separate method.

Reviewer 4 Report

Comments and Suggestions for Authors

Dear Authors, thank you for your review paper. The topic is always current and the data collected in your review study are well organized and discussed. Some concerns about the paper length (especially in the first about about normal function etc etc) and the number of references...these will be decided by assistent and Editor in Chief if they are in accordance with journal guidelines. Thank you for interesting review work. 

Author Response

While we recognize that the manuscript is long (and the revised version is longer), we believe that a comprehensive treatment of this topic is warranted. Moreover, the other reviewers request modifications to expand on the topics rather than decrease the content.

Round 2

Reviewer 2 Report

Comments and Suggestions for Authors

The authors addressed my comments. Accept